# Protective mitochondrial fission induced by stress-responsive protein GJA1-20k

Daisuke Shimura[1], Esther Nuebel[2,3,4], Rachel Baum[1], Steven E Valdez[1], Shaohua Xiao[5], Junco S Warren[1], Joseph A Palatinus[1], TingTing Hong[1,6,7], Jared Rutter[2,3,6], Robin M Shaw[1]*

[1]Nora Eccles Harrison Cardiovascular Research and Training Institute, University of Utah, Salt Lake City, United States; [2]Howard Hughes Medical Institute, University of Utah, Salt Lake City, United States; [3]Department of Biochemistry, University of Utah, Salt Lake City, United States; [4]Biomedical Sciences, Noorda College of Osteopathic Medicine, Provo, United States; [5]Department of Neurology, University of California at Los Angeles, Los Angeles, United States; [6]Diabetes and Metabolism Research Center, University of Utah, Salt Lake City, United States; [7]Department of Pharmacology and Toxicology, College of Pharmacy, University of Utah, Salt Lake City, United States

**Abstract** The Connexin43 gap junction gene *GJA1* has one coding exon, but its mRNA undergoes internal translation to generate N-terminal truncated isoforms of Connexin43 with the predominant isoform being only 20 kDa in size (GJA1-20k). Endogenous GJA1-20k protein is not membrane bound and has been found to increase in response to ischemic stress, localize to mitochondria, and mimic ischemic preconditioning protection in the heart. However, it is not known how GJA1-20k benefits mitochondria to provide this protection. Here, using human cells and mice, we identify that GJA1-20k polymerizes actin around mitochondria which induces focal constriction sites. Mitochondrial fission events occur within about 45 s of GJA1-20k recruitment of actin. Interestingly, GJA1-20k mediated fission is independent of canonical Dynamin-Related Protein 1 (DRP1). We find that GJA1-20k-induced smaller mitochondria have decreased reactive oxygen species (ROS) generation and, in hearts, provide potent protection against ischemia-reperfusion injury. The results indicate that stress responsive internally translated GJA1-20k stabilizes polymerized actin filaments to stimulate non-canonical mitochondrial fission which limits ischemic-reperfusion induced myocardial infarction.

*For correspondence:
Robin.Shaw@hsc.utah.edu

Competing interest: The authors declare that no competing interests exist.

## Introduction

Ischemia-Reperfusion (I/R) injury is known to induce excessive reactive oxygen species (ROS) in mitochondria, which results in cellular dysfunction and organ damage. Interestingly, the phenomenon of ischemic preconditioning, first described 35 years ago (*Murry et al., 1986*), is a potent yet ironic protection of organs from ischemia-induced damage achieved by preceding the full ischemic insult with shorter bouts of ischemia. Despite the large therapeutic potential of preconditioning for any organ such as heart, kidney, skeletal muscle, or brain subjected to anticipated ischemia, the mechanisms of preconditioning are not well understood, nor has an intervention been identified to successfully translate the phenomenon into clinical utility (*Heusch and Gersh, 2020*).

Related to preconditioning is the perplexing dynamic regulation of mitochondria itself. Mitochondria undergo both fission and fusion in an adaptive equilibrium which directly affects cellular activity and response to stress (*Youle and van der Bliek, 2012*; *Friedman and Nunnari, 2014*). It is not clear if an overall shift to fission or fusion is sufficient to define mitochondrial fidelity, or whether a change in the fission-fusion equilibrium occurs secondary to multiple distinct pathways which could be either beneficial or harmful, depending on the pathway taken. For instance, is mitochondrial

fission a causative element of ROS generation, apoptosis, cellular senescence, and cell death (*Suen et al., 2008*; *Wang et al., 2017*; *Nishimura et al., 2018*), or is fission actually protective, such as by promoting mitophagy which can promote survival (*Shirakabe et al., 2016*; *Burman et al., 2017*)? The nuance in understanding the context of mitochondrial fission helps explain the complex relationship between mitochondrial morphology and function (*Nunnari and Suomalainen, 2012*; *Picard et al., 2013*; *Song et al., 2015*). If we have a better understanding of conditions in which mitochondrial fission results from a protective mechanism, we will be closer to learning how to preserve mitochondrial fidelity in the setting of ischemic and reperfusion stress.

Both the gap junction protein Connexin43 (Cx43) and mitochondria are associated with preconditioning protection (*Garcia-Dorado et al., 2006*; *Basheer et al., 2018*; *Rodriguez-Sinovas et al., 2018*). Little is known how the gap junction channel and organelle convey protection during preconditioning. *GJA1*, which encodes Cx43, has a single coding exon and thus is not subject to splicing (*Smyth and Shaw, 2013*). However, *GJA1* mRNA is subject to endogenous internal translation generating several N-terminal truncated isoforms (*Smyth and Shaw, 2013*; *Salat-Canela, 2014*; *Ul-Hussain et al., 2014*). GJA1-20k, which contains the full Cx43 C-terminus but lacks transmembrane domains, is the most abundant and most common smaller isoform and is essential to full-length Cx43 trafficking (*Smyth and Shaw, 2013*; *Xiao et al., 2020*) by recruiting cytoplasmic actin to organize trafficking pathways (*Basheer et al., 2017*). In addition, GJA1-20k is highly enriched at the outer mitochondrial membrane (*Fu et al., 2017*; *Basheer et al., 2018*). GJA1-20k abundance increases with hypoxic and ischemic stress (*Basheer et al., 2018*), resulting in a phenotypically profound cardiac protection that mimics ischemic preconditioning (*Basheer et al., 2018*; *Wang et al., 2019*). It is not known how GJA1-20k protects mitochondria during stress.

In the present study, we found an inverse relationship between the presence of GJA1-20k and mitochondria size in cultured cells and mice. This relationship is not affected by typical mediators of mitochondrial dynamics by Dynamin Related Protein 1 (DRP1), but rather is strongly dependent on actin dynamics. With the generation of smaller actin-associated mitochondria, we observed decreased oxygen consumption, decreased ROS generation, and profound protection against ischemia-reperfusion damage. Our data demonstrate a novel non-canonical mechanism of protective mitochondrial fission which is dependent on GJA1-20k and cytoskeletal dynamics. The findings also identify GJA1-20k as a pharmaceutical candidate for protection of organs undergoing anticipated ischemia.

## Results

### GJA1-20k induces mitochondrial fission in vitro and in vivo

To understand the interplay between GJA1-20k and mitochondria, we expressed GJA1-20k in HEK293 cells and analyzed mitochondrial morphology by fluorescence imaging. GJA1-20k co-localizes with mitochondria and its presence results in a more rounded and punctate appearance of the mitochondria. GJA1-20k also induces a 54 % decrease in average mitochondrial area (*Figure 1A and F*). siRNA-mediated knock-down of *GJA1* expression (*Figure 1—figure supplement 1A*) conversely increases mitochondrial area by 20 % (*Figure 1B and F*). Furthermore, mitochondrial size after *GJA1* knock-down can be rescued by siRNA-resistant GJA1-20k overexpression, but not with overexpression of siRNA resistant Cx43-M6L, which expresses full-length Cx43 without the truncated isoforms (*Smyth and Shaw, 2013*; *Figure 1—figure supplement 1B, C*). Together, these data suggest that the short GJA1-20k isoform, but not full-length Cx43, reduces mitochondrial size.

Since GJA1-20k is abundantly expressed in cardiomyocytes (CMs) suggesting an important function in heart (*Smyth and Shaw, 2013*; *Xiao et al., 2020*), and GJA1-20k can convey ischemic preconditioning protection in heart (*Basheer et al., 2018*) we used cardiomyocytes as a primary cell model to explore GJA1-20k function. We overexpressed GJA1-20k by adenovirus transduction in wild type (WT) neonatal mouse CMs and found a decrease in the average mitochondria area (*Figure 1C and F*). We then used neonatal CMs from a *Gja1^M213L/M213L^* mouse line that was recently generated by mutating the internal AUG (Methionine) at residue 213 of *Gja1* to UUA (Leucine). This M213L mutation removes the internal translation start site, resulting in expression of full length Cx43 protein but not the internally translated GJA1-20k isoform (*Xiao et al., 2020*). Neonatal CMs from *Gja1^M213L/M213L^* mice have an increase in their average area of mitochondria (*Figure 1C and F*), in contrast to the decrease in

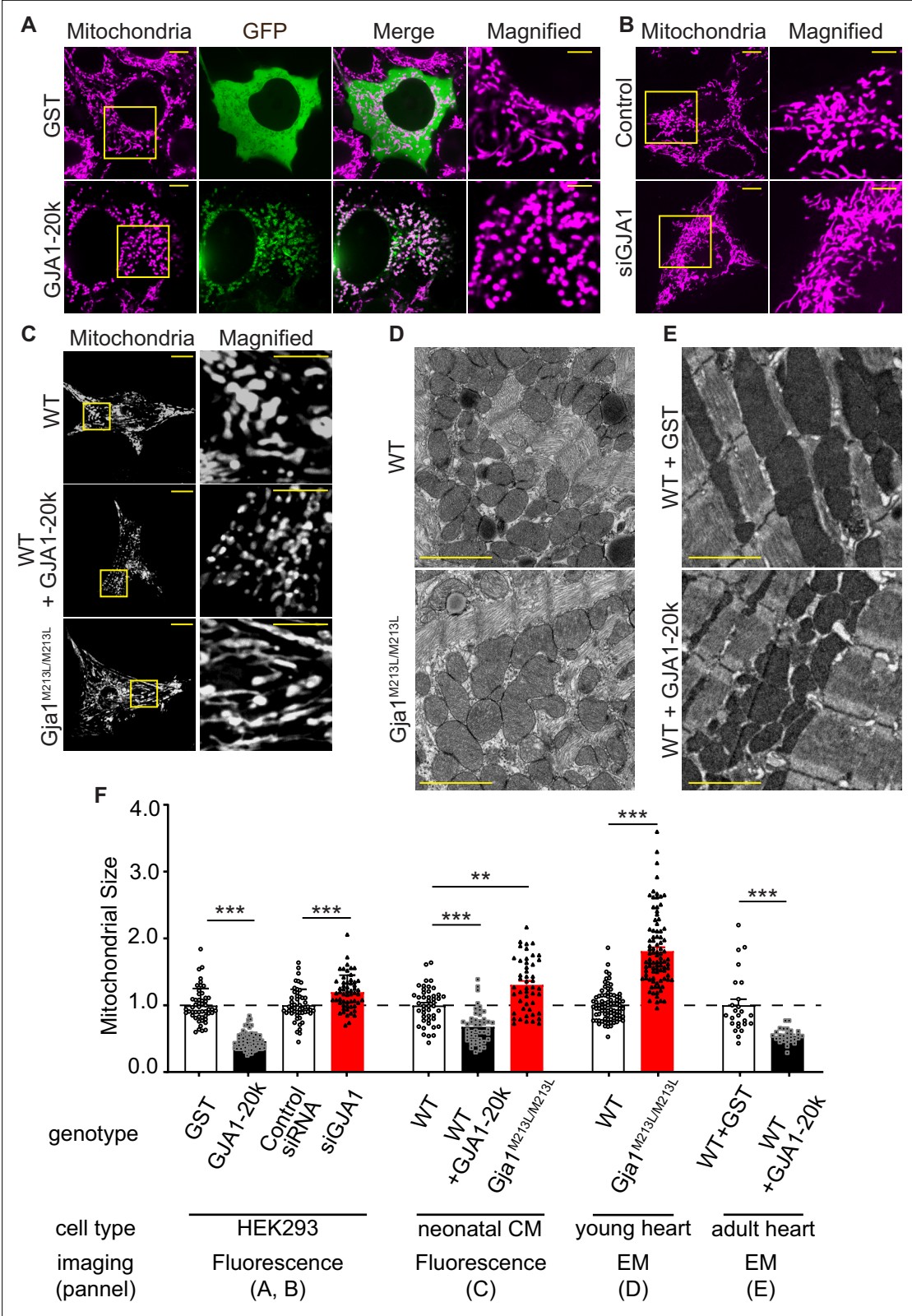

**Figure 1.** GJA1-20k decreases in mitochondrial size. (**A–C**) Representative live cell images of mitochondria in HEK293 (**A**, GST- or GJA1-20k-transfected; **B**, Control and siGja1) and mouse neonatal cardiomyocytes (**C**, WT, GJA1-20k-transduced, and Gja1$^{M213L/M213L}$). The right-most panels are magnified images. (**D**) and (**E**) Representative EM images from young mouse hearts (**D**, WT or Gja1$^{M213L/M213L}$) and adult mouse hearts (**E**, GST- or GJA1-20k-injected). (**F**) Summary of the fold change in the average area of mitochondria. (n = 51 (GST), 67 (GJA1-20k), 52 (Control), or 57 (siGja1) HEK293) cells

*Figure 1 continued on next page*

*Figure 1 continued*

from five independent experiments; n = 46 (WT), 47 (GJA1-20k), or 48 (Gja1^M213L/M213L^) cells from four hearts; n = 84 (WT) or 91 (Gja1^M213L/M213L^) images from six hearts; n = 25 (GST), or 28 (GJA1-20k) images from three hearts. Graphs were expressed as mean ± SD (HEK293) or± SEM (mouse). p values were determined by two-tailed Mann-Whitney U-test or Kruskal-Wallis test with Dunn's post-hoc test. **p < 0.01, ***p < 0.001. Scale bars, 10 μm and 5 μm in magnified (**A–C**); 2 μm (**D and E**). Exact p values and statistical data are provided in the source data.

The online version of this article includes the following source data and figure supplement(s) for figure 1:

**Source data 1.** All data points of the mitochondrial size and the statistical data for *Figure 1F*.

**Figure supplement 1.** The confirmation of Gja1 knock-down and the mitochondrial morphology rescued by GJA1-20k.

**Figure supplement 1—source data 1.** All data points of the mitochondrial size and the statistical data for *Figure 1—figure supplement 1*.

**Figure supplement 2.** The effects of GJA1-20k on membrane potential, ATP production, mitochondrial biogenesis, and mitophagy.

**Figure supplement 2—source data 1.** All data points of the intensity of TMRE, ATP amount, mitochondria DNA copy number, and the protein expression levels and the statistical data for *Figure 1—figure supplement 2*.

mitochondrial area of neonatal CMs with GJA1-20k overexpression (*Figure 1C and F*). Thus, both cultured cells and neonatal mouse cardiomyocytes exhibit an inverse relationship between the presence of GJA1-20k and mitochondrial size.

The homozygous *Gja1^M213L/M213L^* mice, which lack the expression of GJA1-20k but not full-length Cx43, die from poor gap junction trafficking and arrhythmogenic sudden cardiac death two to four weeks after birth (*Xiao et al., 2020*). We therefore pre-emptively sacrificed and examined two-week old homozygous *Gja1^M213L/M213L^* mice, and explored mitochondrial size by electron microscopy (EM). The mitochondria of 2-week-old cardiomyocytes deficient of GJA1-20k are enlarged (*Figure 1D and F*). We then delivered exogenous GJA1-20k using adeno-associated virus type 9 (AAV9) to 8 week old adult WT mice (*Basheer et al., 2018*) and found a decrease in the average mitochondrial area (*Figure 1E and F*). Taken together, in cell lines, neonatal CM, young mouse hearts, and mature mouse hearts, an increase in GJA1-20k results in smaller mitochondria, whereas a decrease in GJA1-20k results in larger mitochondria. Previously, we reported that GJA1-20k is involved in mitochondrial biogenesis (*Basheer et al., 2018*). Consistent with our previous study, AAV9-transduced adult cardiomyocytes showed increased mitochondrial DNA copy number and GJA1-20k deficient mice (Gja1^M213L/M213L^) had decreased copy number. However, exogenous GJA1-20k did not alter the mitochondrial biogenesis in HEK293 cells. Nor did exogenous GJA1-20k affect membrane potential or baseline ATP production (*Figure 1—figure supplement 2A-C*). In addition to mitochondrial DNA copy number, neither biogenesis nor mitophagy protein markers were altered in either GJA1-20k transfected HEK293 cells or Gja1^M213L/M213L^ mouse hearts (*Figure 1—figure supplement 2D-G*).

## Canonical mitochondrial dynamics are not involved

Mitochondrial dynamics are regulated by well-known mediators including DRP1 for fission (*Friedman and Nunnari, 2014*), and MFN1 and MFN2 for fusion (*Schrepfer and Scorrano, 2016*). DRP1 is potentiated by phosphorylation of its Serine 616 and 637 (*Sabouny and Shutt, 2020*). We tested abundance of total DRP1 and phosphorylated DRP1 in GJA1-20k transfected HEK293 cells and found no significant difference in either the protein levels of total DRP1, DRP1-pS616, -pS637, or in the ratio (pS616/total) (*Figure 2A and B*). The levels of MFN1 and MFN2, and mitochondrial marker protein TOM20, a marker of cellular mitochondrial content, were also unchanged (*Figure 2A and B*).

To further investigate whether GJA1-20k induced reduction in mitochondrial size is dependent on DRP1, we analyzed mitochondrial morphology after inhibiting DRP1 by performing siRNA-mediated *DRP1* knock-down (*Figure 2—figure supplement 1A-C*) or transfecting DRP1 dominant negative mutant (K38A), all with or without GJA1-20k transfection. With either method of DRP1 inhibition, the average area of individual mitochondria increased, consistent with inhibiting canonical fission (*Figure 2C and D*). In addition, K38A has more pronounced DRP1 inhibition which resulted in greater mitochondrial enlargement than siDRP1 (*Figure 2C and D*; *Figure 2—figure supplement 1F*). However, GJA1-20k acts epistatically to DRP1 loss or interference and prevents DRP1-mediated mitochondrial enlargement (*Figure 2C–F*; *Figure 2—figure supplement 1B,C*), indicating GJA1-20k can act at or downstream of DRP1 activation. We also interfered mitochondrial dynamics by treating cells pharmacologically with mitochondrial division inhibitor 1 (Mdivi-1) and found same effects of GJA1-20k to rescue the dynamics (*Figure 2—figure supplement 1D,E*). In addition to DRP1,

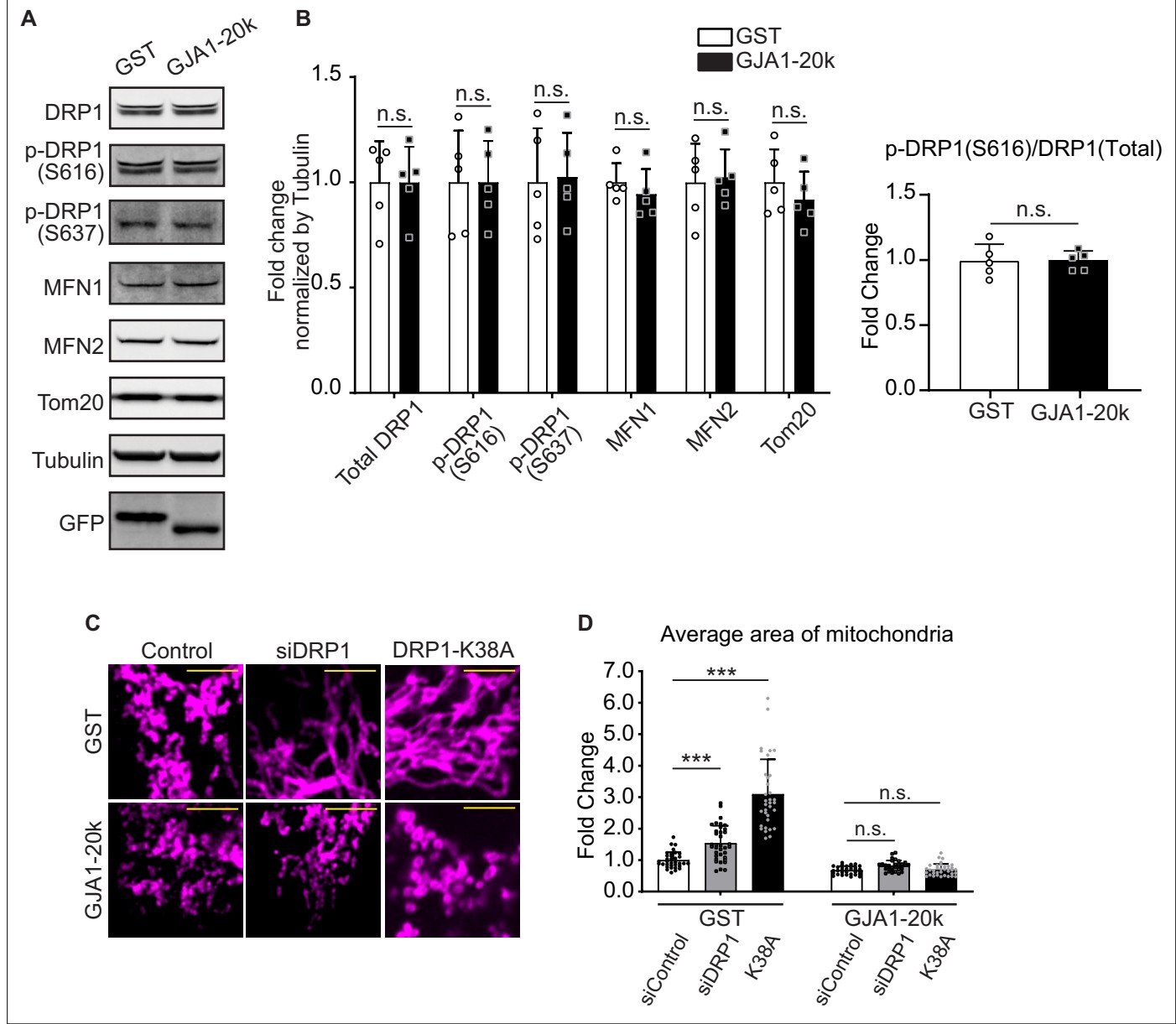

**Figure 2.** DRP1 is not involved in GJA1-20k-mediated mitochondrial fission. (**A**) and (**B**) Western blot analysis for mitochondrial dynamics related proteins. Transfection was confirmed by GFP bands and the band size difference in GFP is due to the difference in molecular weight between GST and GJA1-20k (**A**). Tubulin was used as internal loading control. n = 5 independent experimental repeats. (**C**) Representative fixed cell images of mitochondria (visualized by Tom20) with DRP1 siRNA, K38A treatment, or Control. (**D**) The fold change in the average area of mitochondria in each treatment. n = 34 (GST, control siRNA), 32 (GJA1-20k, control siRNA), 36 (GST, DRP1 siRNA), 31 (GJA1-20k, DRP1 siRNA), 34 (GST, K38A), or 36 (GJa1-20k, K38A) cells from three independent experiments. Graphs were expressed as mean ± SD. p values were determined by two-tailed Mann-Whitney U-test or two-way ANOVA with Bonferroni's post-hoc test. ***$p < 0.001$; n.s., not significant. Scale bars, 5 μm (**C**). Exact p values and statistical data are provided in the source data.

The online version of this article includes the following figure supplement(s) for figure 2:

**Source data 1.** All data points of the protein expression and the mitochondrial size and the statistical data for *Figure 2*.

**Figure supplement 1.** Detailed analysis of DRP1 and DNM2 inhibition and GJA1-20k induced mitochondrial fission.

**Figure supplement 1—source data 1.** All data points of % mitochondria area with DRP1 and mitochondrial size and the statistical data for *Figure 2—figure supplement 1*.

Dynamin-2 (DNM2) is also a GTPase which can potentially regulate mitochondrial fission (*Lee et al., 2016*). To explore possible DNM2 involvement, we knocked-down DNM2 by siRNA and analyzed mitochondrial morphology. In contrast to siDRP1, siDNM2 alone did not affect mitochondrial morphology (*Figure 2—figure supplement 1G-I*), with or without GJA1-20k, suggesting no involvement of DNM2 in mitochondrial dynamics of HEK293 cell lines. It has been reported that DNM2 is not necessary to be responsible for mitochondrial fission in other cell lines as well (*Fonseca et al., 2019*). The fact that DNM2 is not needed for GJA1-20k activity to occur, indicates it is not an essential co-factor in GJA1-20k-mediated mitochondrial fission.

## GJA1-20k interacts with actin to induce fission

The actin cytoskeleton and its dynamics have been implicated as fundamental mediators of mitochondrial fission (*Korobova et al., 2013*; *Hatch et al., 2014*; *Ji et al., 2015*; *Moore et al., 2016*). GJA1-20k has been previously identified to cluster with and stabilize the actin network both in vitro and in vivo (*Basheer et al., 2017*). We imaged actin in GJA1-20k transfected HEK293 cells and noted that, in the presence of GJA1-20k, actin assembles around mitochondria, forming filamentous rings surrounding the outer membrane (*Figure 3A*). We used biochemical methods to confirm that GJA1-20k induces association of actin and mitochondria. Cytosolic and mitochondrial pools of proteins were separated and actin protein levels were measured in each fraction. As seen in *Figure 3B and C*, the large increase in mitochondrial associated actin occurs in the presence of GJA1-20k. Together these data provide both imaging and biochemical evidence that GJA1-20k recruits the actin cytoskeleton to the mitochondrial membrane.

The actin rings of *Figure 3A* are filamentous, indicating formation or stabilization of actin polymers. We asked how GJA1-20k alone, which localizes to the mitochondrial outer membrane (*Basheer et al., 2018*), can recruit and polymerize actin around mitochondria (*Figure 3A*). Using a reductionist cell-free assay of actin polymerization, we found that GJA1-20k does not directly promote actin polymerization (*Figure 3D*), and the polymerization kinetics is even slightly slowed down in the presence of GJA1-20K. However, GJA1-20k causes a leftward shift in the polymerization-depolymerization equilibrium (*Blanchoin et al., 2014*) by directly inhibiting actin depolymerization (*Figure 3E*). The effect of substantial GJA1-20k, by inhibiting depolymerization, is to cause a net increase in polymerization. Because GJA1-20k enriches at outer mitochondrial membrane (*Fu et al., 2017*; *Basheer et al., 2018*), the GJA1-20k-induced polymerization will occur at and around the mitochondrial membrane. Interestingly, GJA1-20k results in net actin polymerization even in the presence of Latrunculin A (LatA), a known potent inhibitor of actin polymerization (*Fujiwara et al., 2018*; *Figure 3E*). In cells, it has been reported that LatA increases individual mitochondrial area, consistent with an inhibition of actin-dependent mitochondrial fission (*Figure 3F and G*; *Korobova et al., 2013*; *Moore et al., 2016*; *Li et al., 2018*). However, the presence of GJA1-20k counteracts the LatA effect, preserving acting polymerization and preventing an overall increase in mitochondrial size (*Figure 3F and G*).

To investigate DRP1 involvement in the actin assemble by GJA1-20k, we observed the actin dynamics under siDRP1 treatment. Even under siDRP1, actin was assembled around mitochondria similarly to *Figure 3A* (*Figure 4A*), suggesting that DRP1 is not necessary to create actin ring around mitochondria by GJA1-20k.

Simultaneous use of fluorescently labelled actin, GJA1-20k, and mitochondria in live cells permit real time imaging of mitochondrial fission events at actin assembly sites. As seen in *Video 1* and *Figure 4B*, GJA1-20k recruits actin to mitochondria, which results in fission. In *Video 1*, the actin network can be seen to develop around mitochondria and, coinciding with GJA1-20k intensity, forms an increasingly tight band across a mitochondrion which, within one minute, results in mitochondrial fission. The imaging in the bottom row of *Figure 4B*, and in the right column of *Video 1* were obtained by multiplying GJA1-20k signal with actin signal, highlighting the locations at which GJA1-20k and actin are coincident. The respective line-scan profiles in *Figure 4C* indicate that mitochondrial fission occurs at points where the product of GJA1-20k and actin is the highest. Following accumulation of GJA1-20k and actin (red lines) at these points, a drop in mitochondrial signal (blue lines) is apparent when fission occurs. Fission (low point of blue lines) occurs approximately 45 seconds after co-accumulation of GJA1-20k and actin (high point of red lines, *Figure 4C*). Time to fission was computed from the time of peak GJA1-20k and actin intensity product, to the time of mitochondrial signal being reduced to background (*Figure 4D–F*). Statistically, this time to fission occurred at a median of

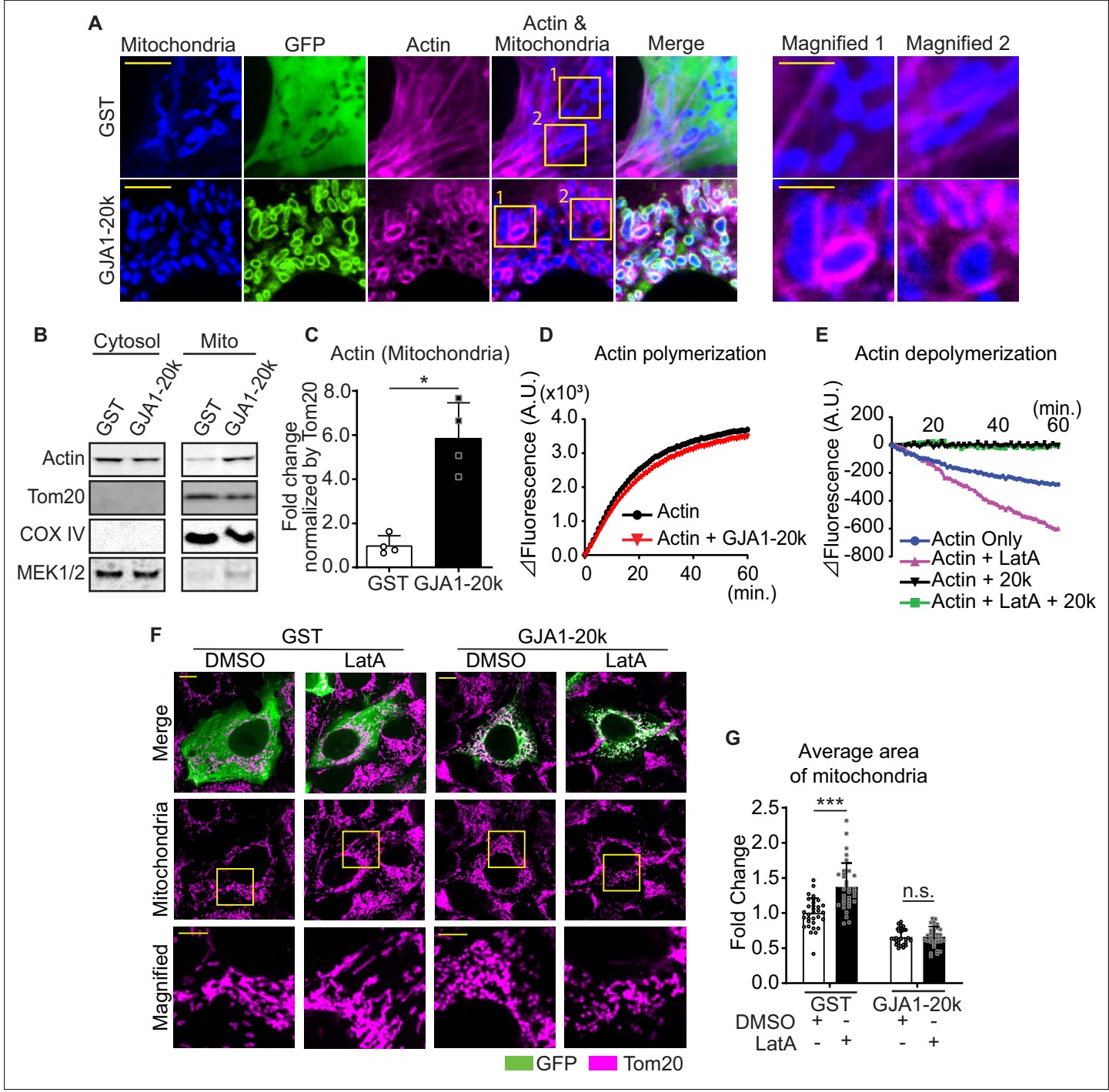

**Figure 3.** GJA1-20k stabilizes and recruits actin around mitochondria for fission. (**A**) Representative live cell images of mitochondria with or without GJA1-20k. Actin was labeled by co-transfection with lifeAct-mCherry. The right-most panels indicate magnified images surrounded by square. (**B**) and (**C**) Western blot analysis in cytosol or mitochondrial fraction. MEK1/2 was used as cytosol marker and Tom20 and COX IV as mitochondrial markers (**B**). Quantification of actin in mitochondrial fraction normalized by Tom20 expression (**C**). n = 4 independent experimental repeats. (**D**) and (**E**) Cell-free actin polymerization (**D**) and depolymerization (**E**) assay. (**F**) Representative fixed cell images of mitochondria (visualized by Tom20) with or without LatA. (**G**) The fold change in the average area of mitochondria with or without LatA treatment. n = 32 (GST, DMSO), 33 (GST, LatA), 28 (GJA1-20k, DMSO), or 32 (GJA1-20k, LatA) cells from three independent experiments. Graphs were expressed as mean ± SD. p values were determined by two-tailed Mann-Whitney U-test or two-way ANOVA with Bonferroni's post-hoc test. *p < 0.05, ***p < 0.001; n.s., not significant. Scale bars, 10 μm (**F**), 5 μm (**A** and magnified in **F**) and 2 μm (magnified in **A**). Exact p values and statistical data are provided in the source data.

The online version of this article includes the following figure supplement(s) for figure 3:

**Source data 1.** All data points of the actin assay and the mitochondrial size and the statistical data for *Figure 3*.

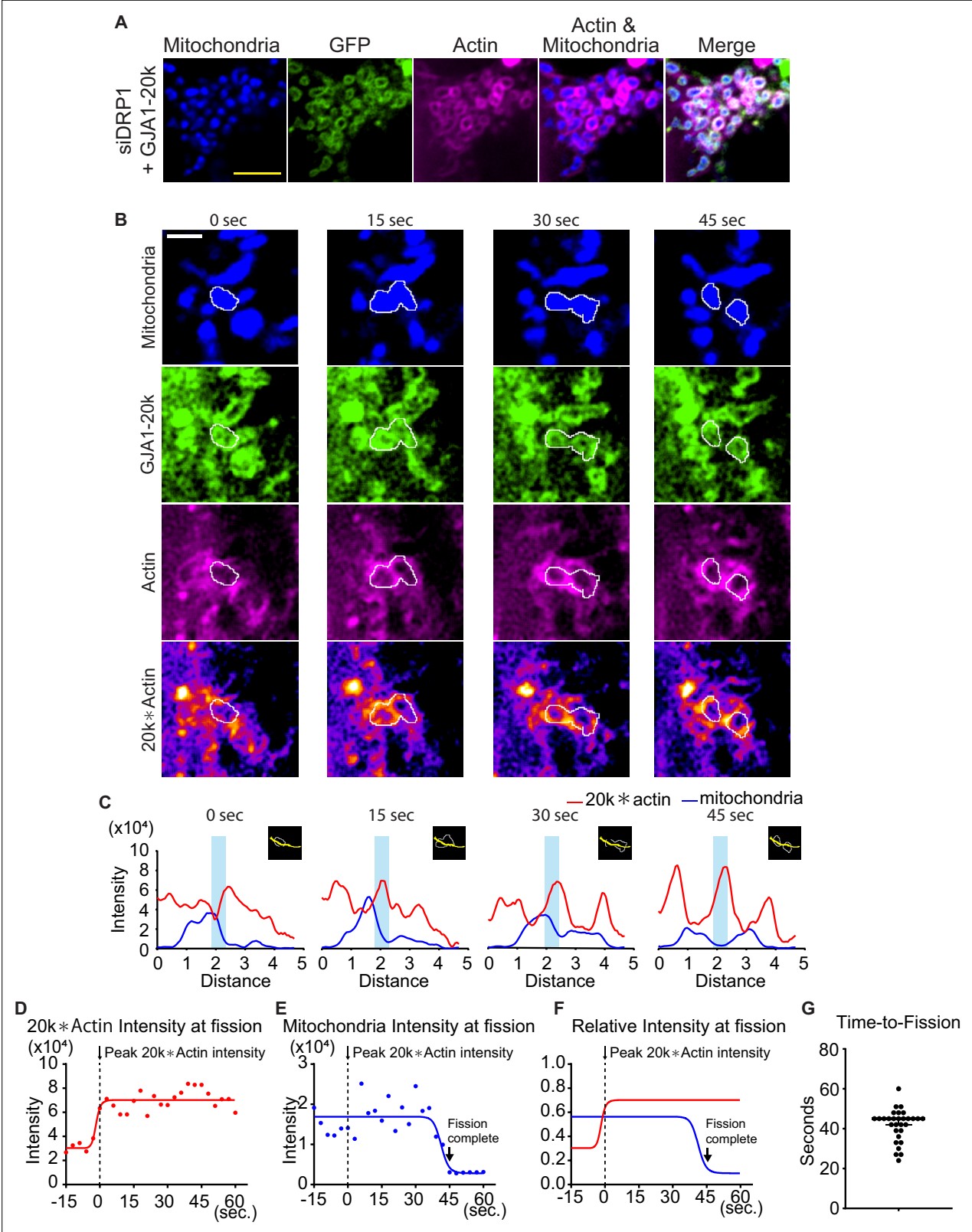

**Figure 4.** Time course of mitochondria dynamics and actin accumulation at the mitochondrial fission site under siDRP1. (**A**) Representative live cell images of mitochondria with GJA1-20k under siDRP1. (**B**) Representative mitochondrial dynamics in GJA1-20k-transfected cells by time-lapse live cell imaging under siDRP1 treatment. The bottom images (with fire look-up table) indicate the product of GJA1-20k and actin signals. The white lines indicate mitochondrial outlines. Scale bars, 5 μm (**A**) and 2 μm (**B**). (**C**) The intensity of mitochondria and the product of GJA1-20k and actin from (**B**). The

*Figure 4 continued on next page*

*Figure 4 continued*

colored areas indicate fission sites along the lines shown in each respective insert. (**D–F**) The time course of the product of GJA1-20k and actin intensity (**D**), mitochondrial intensity (**E**), or combined relative intensity (**F**) at the fission site from the mitochondrion shown in (**B**). Curves are four parameters logistic (4PL) fits to the data. Time 0 corresponds to the peak product of actin and GJA1-20k intensity. The arrows indicate the time point at which fission is complete. (**G**) Measured time from peak product of actin and GJA1-20k intensity to fission in seconds (bars indicate median and 95 % confidence interval). n = 30 events. All data points are provided in the source data.

The online version of this article includes the following figure supplement(s) for figure 4:

**Source data 1.** All data points of the intensity and time-to-fission for *Figure 4*.

**Figure supplement 1.** Time course of mitochondria dynamics and actin acumulation at the mitochondrial fission site under Mdivi-1 treatment.

**Figure supplement 1—source data 1.** All data points of All data points of the intensity and time-to-fission for *Figure 4—figure supplement 1*.

45 seconds, with a standard deviation of 11 seconds (*Figure 4G*). Note, the real time imaging shown in *Video 1*, and *Figure 4* were performed under siDRP1. Therefore, the mitochondrial fission induced by cooperation between GJA1-20k and actin can be independent of canonical DRP1-mediated fission. To rule out inadvertent bias by siRNA, we used pharmacologic Mdivi-1 to inhibit DRP1 and, similar to the use of DRP1 siRNA, actin formed around mitochondria at GJA1-20k sites (*Figure 4—figure supplement 1A-D*) and fission occurred within a similar timescale (*Videos 2 and 3*; *Figure 4—figure supplement 1E-H*).

## GJA1-20k induces protective effects against oxidative stress

It has been found that AAV9-mediated GJA1-20k overexpression in mouse heart depresses mitochondrial respiration, which has the beneficial effect of limiting myocardial infarction after ischemic-reperfusion (I/R) injury (*Basheer et al., 2018*). We measured oxygen consumption rate (OCR) using a Seahorse mitochondrial stress test in GJA1-20k transfected HEK293 cells and found a decrease in maximal respiration (*Figure 5A and B*), consistent with previous results (*Basheer et al., 2018*). Similarly, maximal respiration is increased in neonatal CMs derived from GJA1-20k deficient $Gja1^{M213L/M213L}$ mice and maximal respiration for heterozygous $Gja1^{M213L/WT}$ mice is between that of WT and $Gja1^{M213L/M213L}$ (*Figure 5C and D*; *Figure 5—figure supplement 1A,B*). In addition, observing other OCR parameters, we found a decrease in ATP-linked respiration and reserve capacity in $Gja1^{M213L/WT}$ cardiomyocytes, and an increase in proton leak

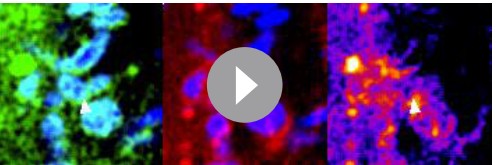

**Video 1.** GJA1-20k assembles actin to mitochondrial fission sites and causes fission under siDRP1. The mitochondrial fission in GJA1-20k-transfected HEK293 cells was monitored under siDRP1 treatment. GJA1-20k (tagged with GFP, green) and mitochondria (indicated by Mitotracker, blue) were simultaneously imaged (left panel), as were mitochondria and actin (indicated by LifeAct-mCherry, red) (middle panel). The spatial coincidence of GJA1-20k and actin were obtained by multiplying GJA1-20k signal with mitochondrial signal, as indicate with a polychromatic fire lookup table (right panel). Note GJA1-20k then actin surround the mitochondria, assembles at the neck of fission site, resulting in fission. The white arrowhead indicates the fission point. Images for each timepoint were obtained every 3 s. There are 17 images for a 51 second clip, played back at five frames per second (about 15 times as fast as real time). Still frames are indicated in Figure 4B.

https://elifesciences.org/articles/69207/figures#video1

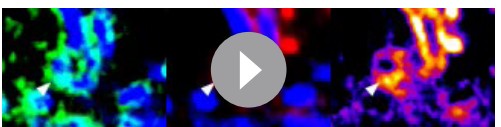

**Video 2.** GJA1-20k assembles actin to mitochondrial fission sites and causes fission under Mdivi-1 (clip 1). The mitochondrial fission in GJA1-20k-transfected HEK293 cells was monitored under Mdivi-1 treatment. GJA1-20k (tagged with GFP, green) and mitochondria (indicated by Mitotracker, blue) were simultaneously imaged (left panel), as were mitochondria and actin (indicated by LifeAct-mCherry, red) (middle panel). The spatial coincidence of GJA1-20k and actin were obtained by multiplying GJA1-20k signal with mitochondrial signal, as indicate with a polychromatic fire lookup table (right panel). Note GJA1-20k then actin surround the mitochondria, assembles at the neck of fission site, resulting in fission. The white arrowhead indicates the fission point. Images for each timepoint were obtained every 3 ss. There are 26 images for a 78 second clip, played back at five frames per second (about 15 times as fast as real time). Still frames are indicated in Figure 4—figure supplement 1A.

https://elifesciences.org/articles/69207/figures#video2

**Video 3.** GJA1-20k assembles actin to mitochondrial fission sites and causes fission under Mdivi-1 (clip 2). The mitochondrial fission in GJA1-20k-transfected HEK293 cells was monitored under Mdivi-1 treatment. GJA1-20k (tagged with GFP, green) and mitochondria (indicated by Mitotracker, blue) were simultaneously imaged (left panel), as were mitochondria and actin (indicated by LifeAct-mCherry, red) (middle panel). The spatial coincidence of GJA1-20k and actin were obtained by multiplying GJA1-20k signal with mitochondrial signal, as indicate with a polychromatic fire lookup table (right panel). Note GJA1-20k then actin surround the mitochondria, assembles at the neck of fission site, resulting in fission. The white arrowhead indicates the fission point. Images for each timepoint were obtained every 3 s. There are 81 images for a 243 second clip, played back at five frames per second (about 15 times as fast as real time). Still frames are indicated in Figure 4—figure supplement 1C.
https://elifesciences.org/articles/69207/figures#video3

and non-mitochondrial respiration in $Gja1^{M213L/M213L}$ suggesting that there can be compensatory long-term effects of the $Gja1$ mutation (*Figure 5—figure supplement 1C*).

To explore whether GJA1-20k is necessary to protect against oxidative injury in heart muscle, we subjected Langendorff-perfused adult hearts from heterozygous $Gja1^{M213L/WT}$ mice to ischemia-reperfusion (I/R) injury. Since homozygous $Gja1^{M213L/M213L}$ mice die in 2–4 weeks after birth, we used heterozygous $Gja1^{M213L/WT}$ mice to be able to work with adult hearts. At baseline, adult $Gja1^{M213L/WT}$ mice with an approximately 50 % reduction of GJA1-20k expression, have no change in overall phenotype, or cardiac chamber dimensions or cardiac functional characteristics compared to WT mice, as measured by echocardiography (*Xiao et al., 2020*). However, remarkably, reduced GJA1-20k expression results in an almost complete cardiac infarction after I/R injury (*Figure 5E and F*). Moreover, ROS production after I/R injury was increased in $Gja1^{M213L/WT}$ mice compared to WT post-I/R (*Figure 5G and H*). There was no significant difference in mitochondria size at the basal condition between WT and $Gja1^{M213L/WT}$ mice adult CMs as with neonatal CMs (*Figure 5I and J*), whereas the mitochondria size was significantly increased after I/R injury and the heterozygous $Gja1^{M213L/WT}$ mice had larger mitochondria compared to WT mice post-I/R (*Figure 5I and J*). Interestingly, the area of mitochondrial matrix was also increased, suggesting loss of cristae in $Gja1^{M213L/WT}$ mice heart (*Figure 5K and L*). These data indicate that even partial deletion of GJA1-20k results in a profoundly impaired response to ischemic stress.

Given that GJA1-20k is a stress-responsive protein which can mediate ischemic preconditioning (*Basheer et al., 2018*) and to explore the protective effects of GJA1-20k against oxidative stress, we measured mitochondrial generation of reactive oxygen species (ROS) in HEK293 cells under oxidative stress induced by exogenous $H_2O_2$ treatment. In unstressed cells, the presence of GJA1-20k does not affect mitochondrial ROS generation (*Figure 5—figure supplement 1D,E*) whereas in the presence of exogenous $H_2O_2$, GJA1-20k prevents an increase in mitochondrial ROS (*Figure 5—figure supplement 1D,E*). Thus, GJA1-20k, which depresses mitochondrial respiration in unstressed cells, does not affect homeostatic ROS generation, yet has the beneficial effect of decreasing toxic ROS generation during stress (*Figure 5A and B*; *Figure 5—figure supplement 1D,E*). To clarify whether the protective effects come from GJA1-20k expression or mitochondrial size, we measured ROS from HEK293 cells with DRP1 overexpression instead of GJA1-20k overexpression. DRP1 overexpression decreased in mitochondrial size similar to GJA1-20k (*Figure 5—figure supplement 1F,G*). However, ROS generation induced by $H_2O_2$ was significantly increased and higher than GST control under $H_2O_2$ (*Figure 5—figure supplement 1H,I*). Taken together, GJA1-20k itself is necessary for the protective mitochondrial fission.

## Discussion

In conclusion, we find that the stress-responsive protein GJA1-20k organizes actin around mitochondria, promoting an actin-dependent mitochondrial fission process that also protects against oxidative and ischemic stress (cartoon in *Figure 6*). This protective fission pathway is distinct from canonical DRP1-mediated fission. In heart muscle, GJA1-20k appears to be critical to limit ischemic damage. Adult mice heterozygous for the internal GJA1-20k start have normal baseline cardiac phenotypes, yet suffer almost complete myocardial infarctions once subjected to I/R injury (*Figure 5E and F*).

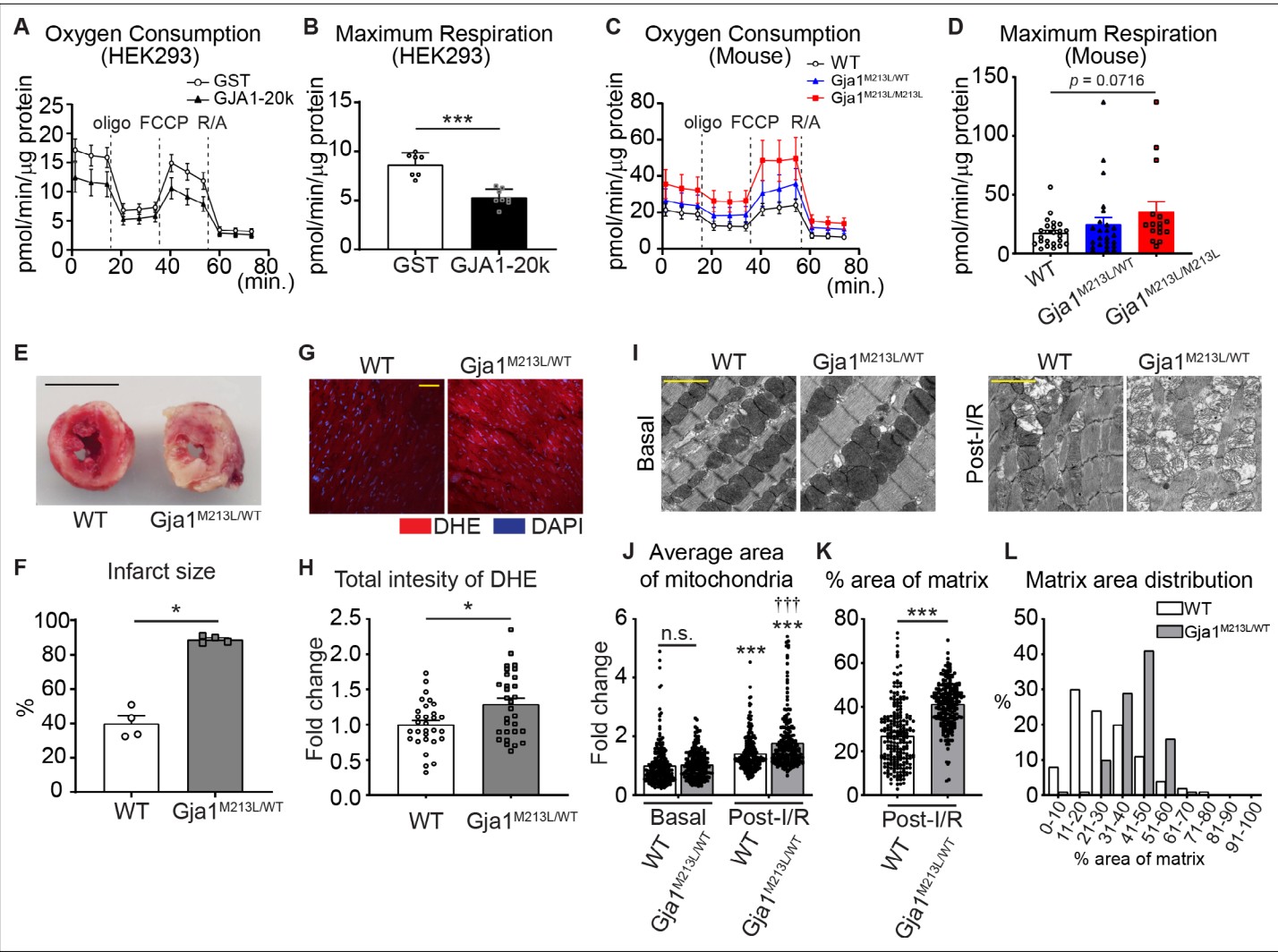

**Figure 5.** Mitochondrial metabolic function is preserved by GJA1-20k. (**A–D**) Real-time change in OCR by Seahorse assay and the maximum respiration in HEK293 cells (**A and B**) and mouse neonatal CM (**C and D**). 7 (GST) or 8 (GJA1-20k) replicates from HEK293 cells; n = 24 (WT and Gja1^M213L/WT from four hearts) or 16 (Gja1^M213L/M213L from three hearts) replicates. (**E**) Representative images of TTC stained hearts from WT and Gja1^M213L/WT mice post-I/R. (**F**) Quantification of infarct size after I/R. n = 4. (**G**) Representative images of DHE staining post-I/R from WT or Gja1^M213L/WT heart. (**H**) Relative intensity of DHE. n = 30 images from three hearts in each genotypes. (**I**) The representative electron microscope images from adult mouse hearts under basal or post-I/R (WT or Gja1^M213L/WT). (**J**) The fold change in the average area of individual mitochondria. n = 240 (WT, Basal), 225 (Gja1^M213L/WT, Basal), 229 (WT, post-I/R), or 224 (Gja1^M213L/WT, post-I/R) mitochondria from three hearts. (**K**) % area of mitochondria matrix post-I/R. n = 229 (WT, post-I/R), or 224 (Gja1^M213L/WT, post-I/R) mitochondria from three hearts. (**L**) The distribution of the data set in (**K**). Graphs were expressed as mean ± SD (HEK293) or SEM (mouse). p values were determined by two-tailed Mann-Whitney U-test, Kruskal-Wallis test with Dunn's post-hoc test, or two-way ANOVA with Bonferroni's post-hoc test. *p < 0.05, ***p < 0.001; †††p < 0.001 compared to WT post-I/R; n.s., not significant. Scale bars, 5 mm (**E**); 50 μm (**G**); 2 μm (**I**). Exact p values and statistical data are provided in the source data.

The online version of this article includes the following figure supplement(s) for figure 5:

**Source data 1.** All data points of the seahorse assay, infarct size, DHE intensity, and mitochondrial analysis and the statistical data for *Figure 5*.

**Figure supplement 1.** The details of difference between WT and heterozygous Gja1^M213L/WT and ROS generation induced by $H_2O_2$ stimulation.

**Figure supplement 1—source data 1.** All data points of the mitochondrial size, the detailed seahorse data, and intensity of ROS measurement and the statistical data for *Figure 5—figure supplement 1*.

Mitochondrial fusion and fission dynamics are understood to involve several key molecules. MFN1 and MFN2 have important roles in fusion (*Schrepfer and Scorrano, 2016*), while DRP1 mediates canonical fission (*Friedman and Nunnari, 2014*). A dynamic equilibrium between fusion and fission is important during development as well as in response to a changing cellular environment. Alteration

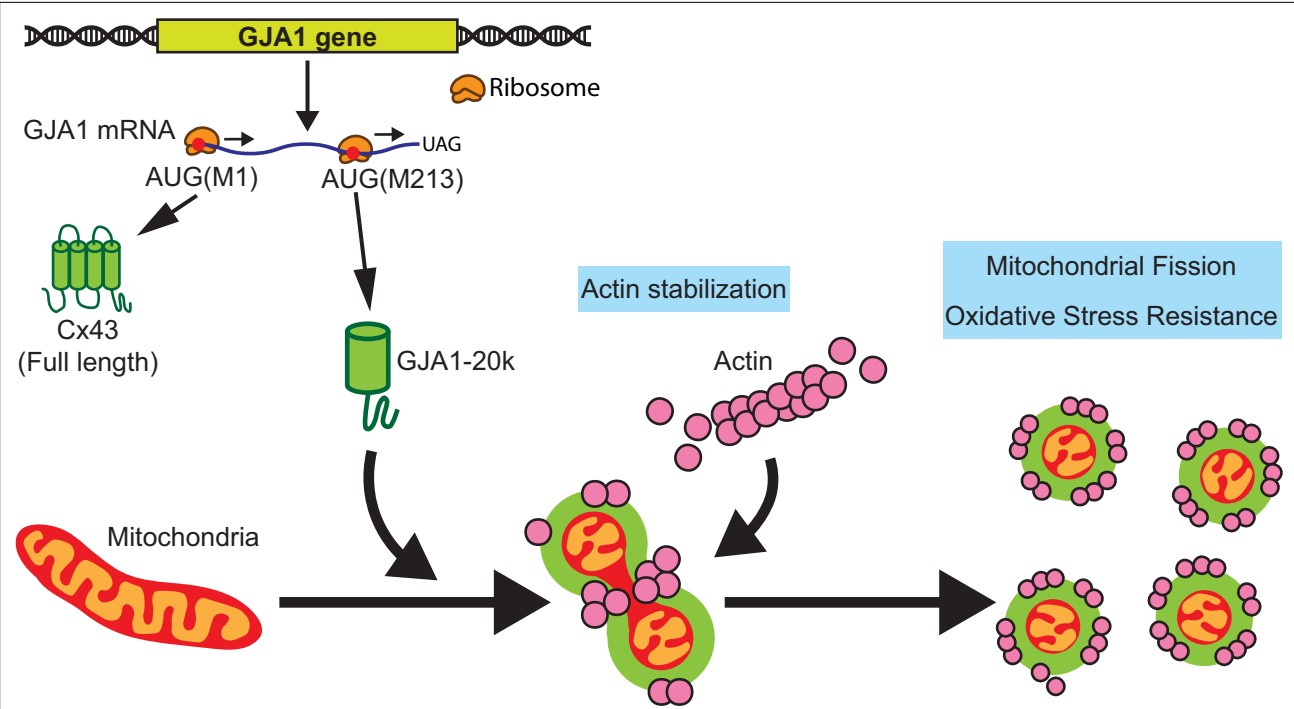

**Figure 6.** Schematic summary. GJA1-20k, internally translated from Gja1 mRNA, localizes mitochondria membrane, stabilizes actin cytoskeleton, and recruits actin around mitochondria to both induce mitochondrial fission and achieve oxidative stress resistance.

of the equilibrium point between fusion and fission occurs with disease (*Chan, 2006*). Additionally, in mouse heart, hypertrophy or dilated cardiomyopathy result from either MFN1 or MFN2 deletion (which causes mitochondrial fragmentation) or DRP1 deletion (which causes hyperfused mitochondria) (*Song et al., 2015*). The implication is that either a leftward or rightward shift in the fusion-fission equilibrium can negatively affect cardiac function. However, it is becoming increasingly clear that mitochondrial size alone, the easiest readout of mitochondrial dynamics, is insufficient to interpret whether destructive or pro-survival pathways are dominant. The presence of GJA1-20k, which increases with ischemic and oxidative stress (*Ul-Hussain et al., 2014*; *Basheer et al., 2018*), is clearly beneficial in the setting of I/R injury (*Figure 5G and H*). Yet the presence of GJA1-20k, while inducing mitochondrial fission and smaller mitochondria (*Figures 1, 3 and 4*), does not either reduce MFN1 or MFN2, activate DRP1, change membrane potential, ATP production, mitochondrial biogenesis, or mitophagy (*Figure 2*; *Figure 1—figure supplement 2*). The ability of GJA1-20k to recruit actin to induce fission independently of DRP1 is novel. GJA1-20k use of actin to cause fission occurs rapidly, within 45 s of peak GJA1-20k and actin coincidence at mitochondrial membrane (*Figure 4G*). We have also investigated the involvement of DNM2 as another GTPase for the mitochondrial fission, but DNM2 knockdown failed to induce sufficient mitochondrial fusion or alter the effects of GJA1-20k (*Figure 2—figure supplement 1G-I*). Therefore, it does not appear that DNM2 is essential for GJA1-20k activity on mitochondrial fission. In addition to actin, the endoplasmic reticulum (ER) membrane can be involved in mitochondrial scission (*Friedman et al., 2011*; *Tandler et al., 2018*). Future studies should be considered whether GJA1-20k induced actin cytoskeleton arrangements involves ER membrane as well.

Given the rapidity (less than one minute) of GJA1-20k induced fission, it is of interest how fission activity can be limited in stable cells, especially terminally differentiated cells such as adult cardiomyocytes. We have not yet evaluated whether secondary regulators or post-translational modification of either primary actin or GJA1-20k affects GJA1-20k mediated mitochondrial dynamics. One simple form of regulation is in the abundance of GJA1-20k. The paucity of GJA1-20k at baseline, and its large and rapid increase during the onset of ischemic stress (*Basheer et al., 2018*), support the role of GJA1-20k as a stress responsive process which affords acute protection during ischemic injury.

While smaller mitochondrial size has been traditionally associated with mitochondrial fragmentation and poor cellular health (*Wai and Langer, 2016*), it is now understood that there is a more complex relationship between mitochondrial size and mitochondrial function (*Sprenger and Langer, 2019*). Smaller mitochondria can signal not only disease but also tissue protection and adaptation. DRP1-induced fission and smaller mitochondria are associated with excess ROS production, apoptosis, and organ injury (*Hu et al., 2017*; *Wang et al., 2017*), which is also seen in *Figure 5—figure supplement 1F-I*, whereas DRP1-independent mitochondrial fission has been observed in cells undergoing protective mitophagy (*Stavru et al., 2013*; *Yamashita et al., 2016*; *Coronado et al., 2018*). It appears that GJA1-20k-mediated fission is highly beneficial, and part of a pro-survival cellular response to stress. Our results suggest that mitochondrial morphology (i.e. the overall balance of fusion and fission) in stressed cells should be considered in the context of the presence of specific proteins. For instance, pathological fission in the presence of activation of DRP1 might indicate mitophagy (*Ikeda et al., 2015*; *Burman et al., 2017*; *Wang et al., 2017*), whereas an increase in GJA1-20k implies a stress-induced reduction of ROS generation and beneficial depression of metabolism. Both DRP1 activation and GJA1-20k generation result in mitochondrial fission, but have different implications for cellular health.

Since its first report 35 years ago, investigators have both tried to understand the mechanisms of preconditioning as well as use putative mediators of precondition to benefit organs undergoing anticipated ischemia. These efforts have included dozens of preclinical and proof-of-concept studies, yet without success (*Heusch and Gersh, 2020*). Mediators such as cyclophilin D (*Cung et al., 2015*), cardiolipin (*Gibson et al., 2016*), mitochondrial permeability transition pore (MPTP)(*Schaller et al., 2010*; *Atar et al., 2015*), Cx43 (*Schulz et al., 2015*), ATP-dependent K$^+$ channels (*Heinzel et al., 2005*; *Garlid et al., 2009*), STAT3 (*Heusch et al., 2011*), GSK3-beta (*Juhaszova et al., 2004*), and opioid receptors *Dragasis et al., 2013* have been implicated in preconditioning protection (*Heusch and Gersh, 2017*), but replicating their involvement in clinical studies has, to present, failed to be successful. More recently, the phenomenon of remote preconditioning has been explored as a therapeutic solution, but clinical application of this approach has also yet to be successful (*Heusch and Gersh, 2020*). It is possible that the reason a central mediator of preconditioning has not been determined is that it has not been available for study. GJA1-20k was only first reported to occur endogenously in 2013 (*Smyth and Shaw, 2013*) and its association with mitochondria and potential beneficial effects for survival of I/R was reported in the last 3 years (*Fu et al., 2017*; *Basheer et al., 2018*; *Wang et al., 2019*; *Fu et al., 2020*; *Ren et al., 2020*). As a smaller isoform of Cx43 that contains the epitope of most anti-Cx43 antibodies, and is localized to mitochondria (*Fu et al., 2017*), GJA1-20k could be central to the studies that implicated Cx43 as a mediator of preconditioning. As a stress-responsive protein (*Ul-Hussain et al., 2014*; *Basheer et al., 2018*), endogenous GJA1-20k is induced by short bouts of ischemia prior to a longer ischemic period (*Basheer et al., 2018*). Because GJA1-20k-induced fission is associated with less ROS production with oxidative stress (*Figure 5—figure supplement 1D*, E), the endogenous generation of GJA1-20k and subsequent decreased ROS production could explain a major benefit of pre-conditioning. Of note, genetic GJA1-20k reduction increases infarct size and ROS production post-I/R injury (*Figure 5E–H*). In addition, the population of damaged mitochondria is significantly increased in heterozygous *Gja1$^{M213L/WT}$* mouse heart post-I/R (*Figure 5I–L*). Therefore, GJA1-20k induced decreases in ROS production could limit the amount of I/R injury induced by myocardial infarction.

Major cardiovascular procedures are associated with low flow and ischemia related damage to end-organs such as the heart, kidneys, and brain. A priori gene therapy mediated delivery of GJA1-20k to these organs could be beneficial. For instance, percutaneous coronary interventions are associated with both cardiac and renal damage (*Brown et al., 2008*; *Tricoci et al., 2018*) and could benefit from GJA1-20k-induced protection. Similarly, coronary bypass surgery with cardiopulmonary bypass is associated with low flow damage to the brain and kidneys (*Roach et al., 1996*; *McKhann et al., 2006*; *O'Neal and Shaw, 2016*), and abdominal aorta surgery is associated with renal damage, all of which may benefit from viral-mediated GJA1-20k pretreatment.

It is also possible, as suggested by unpublished preliminary studies by our group, that GJA1-20k peptide administration intravenously can gain access to the intracellular environments of the heart and other organs. In this scenario, GJA1-20k administration could occur minutes to hours ahead of an ischemic insult rather than the presumably days to weeks which would be needed for viral-mediated

introduction. Future studies will focus on acute intravenous administration of GJA1-20k peptide in subjects about to undergo anticipated ischemia.

The phenomenon of stunned myocardium is common in patients with I/R injury such as experiencing an occluded coronary artery that is revascularized, but the mechanism of stunning is not well elucidated. It is possible that stunned myocardium has elevations of GJA1-20k after the ischemic injury. Of note, stunned myocardium is metabolically quiescent and also exhibits smaller mitochondria (*Borgers et al., 1993*; *Borgers, 2002*), further supporting the possibility that GJA1-20k is involved in the stunned response to ischemia as a mechanism of preventing myocardial death. Endogenous induction of GJA1-20k may also explain other instances when smaller mitochondrial size correlates with a beneficial effect (*Coronado et al., 2018*). Recent studies have identified that exogenous GJA1-20k can protect neurons subjected to traumatic brain injury (*Ren et al., 2020*), hearts from ischemia (*Basheer et al., 2018*) and angiotensin induced hypertrophy (*Fu et al., 2020*). Furthermore, by mimicking the protection afforded by ischemic preconditioning, exogenous GJA1-20k is a promising therapeutic to protect hearts, brains and other organs against expected ischemic damage.

We have found that neither exogenous GJA1-20k overexpression in mouse hearts nor GJA1-20k$^{M213L/WT}$ adult mice have altered unstressed cardiac function (*Basheer et al., 2018*; *Xiao et al., 2020*). However acute administration of GJA1-20k or using GJA1-20k in the context of stressed myocardium could conceivably result in diminished cardiac function. As GJA1-20k is stress responsive, an increase in GJA1-20k may be a mediator of post-ischemic protection (*Basheer et al., 2018*), and even responsible for the phenomenon of myocardial stunning which happens to be associated with small mitochondria (*Borgers et al., 1993*; *Borgers, 2002*). On the other hand, GJA1-20k induced functional hibernation of stressed myocardium, while increasing survival, could conceivably result in diminished cardiac function. If indeed GJA1-20k induces an acute cardiac depressant response, appropriate steps would be needed to compensate for lower cardiac output during the myocardial protection period.

In summary, we identify that upregulation of the stress-responsive internally translated peptide, GJA1-20k, may be a critical mediator of ischemic-preconditioning protection. GJA1-20k induces fission by recruiting actin to mitochondria, inducing a fission that decreases ROS generation and protects organs such as the heart. Use of exogenous GJA1-20k as a therapeutic can potentially realize the long sought yet still elusive clinical need of preventing organs from undergoing damage during anticipated ischemia.

## Materials and methods
### Animals
All mice were maintained under sterile barrier conditions. For the exogenous gene delivery, we used C57BL/6 male mice under same conditions at the age of 8 weeks to start the study procedure according to previous study (*Basheer et al., 2017*; *Basheer et al., 2018*). We injected 100 µl of 3 × 10$^{11}$ vector genomes per mL of Adeno-associated virus type 9 (AAV9) vectors containing GFP-tagged glutathione S-transferase (GST-GFP) or GJA1-20k (GJA1-20k-GFP) driven by the cytomegalovirus (CMV) promoter through retro-orbital injection. Eight weeks post-injection, the heart dissection was performed under anesthesia by isoflurane. Heparin (100 IU, i.p.) was injected 20–30 min before dissection. We perfused the heart with cold HEPES buffer to wash out the blood and immediately freeze and proceed DNA extraction. The details of *Gja1*$^{M213L/M213L}$ mouse generation has been described previously (*Xiao et al., 2020*). We isolated neonatal cardiomyocyte (postnataTl 2–3 days) from *Gja1*$^{M213L/M213L}$ and Wildtype (WT) mice using Pierce Primary Cardiomyocyte Isolation Kit (Thermo Fisher Scientific, Walthman, MA) following manufacturer protocol. The neonatal cardiomyocyte was seeded into gelatin/fibronectin coated 35 mm glass-bottomed dish and subjected to imaging as described below. Adenovirus encoding GJA1-20k-V5 (GFP-V5 as a control; four plaque-forming unit/cell) was transduced as previously described *Basheer et al., 2017* followed by live cell imaging. The average area of individual mitochondria in GFP-V5 control myocyte was unchanged compared to WT non-transduced cardiomyocytes (*Figure 1—figure supplement 1D, E*). We also dissected the heart tissue at the age of 2 weeks from *Gja1*$^{M213L/M213L}$ and WT mice under anesthesia described above for the following experiments. All animal care and study protocols were approved by University of Utah Institutional Animal Care and Use Committee.

## Electron microscope imaging

The mouse hearts were prepared as described previously (*Basheer et al., 2018*). Briefly, the adult mouse hearts were fixed by perfusing with 2 % glutaraldehyde and 2 % paraformaldehyde in PBS for 10 min followed by post-fixed with 1 % osmium tetroxide and incubated in 3 % uranyl acetate. The small hearts from young mouse (2 weeks old) dissected into 1 mm pieces were immediately fixed with 2.5 % glutaraledehyde, 1 % paraformaldehyde, 0.1 M Cacodylate buffer, pH 7.4, 6 mM $CaCl_2$, 4.8 % Sucrose, at 4 °C. Following overnight fixation, the specimens were rinsed two times in buffer and were postfixed in 2 % Osmium tetroxide for 1 hr at room temperature. The specimens were rinsed in $dH_2O$ and pre-stained with uranyl acetate for 1 hr ate room temperature. Then the samples were dehydrated in graded ethanol series and three times in pure acetone then infiltrated and embedded in epoxy resin, Embed 812 (cat # 14121, Electron Microscopy Sciences, Hatfield, PA). The blocks were cut at 70 nm thickness using an ultramicrotome (Leica, Wetzlar, Germany) and poststained with uranyl acetate for 10 min, and lead citrate for 5 min. Sections were examined at an accelerating voltage of 120 kV in a JEM-1400 plus or JEM1200-EX (JEOL, Tokyo, Japan) transmission electron microscope with CCD Gatan camera. The number and the total area of mitochondria in each image were measured using imageJ and the average area was calculated by dividing the total area by the number. Electron microscope imaging was performed by the core facility at Electron Microscopy Laboratory at University of Utah and at the Electron Imaging Center of The California NanoSystems Institute at University of California, Los Angeles.

## Cell culture, plasmid and siRNA transfection, DRP1 inhibition, and Latrunculin a treatment

HEK293 cells were purchased from Thermo Fisher Scientific (R70007) and cultured with DMEM containing 10 % fetal bovine serum (FBS), non-essential amino acids, sodium pyruvate (Thermo Fisher Scientific), and Mycozap-CL (Lonza) in 37 °C, 5 % $CO_2$ incubator. The cell line was tested negative for mycoplasma contamination. For imaging analysis, we coated 35 mm glass-bottomed dish with 0.1 % gelatin (Sigma-Aldrich, St. Louis, MO) and human fibronectin (20 µg/ml, Corning) incubating 37 °C for 2 hr or 4 °C overnight before cell seeding. We seeded the cells ($2.0 \times 10^5$ cells/dish) into the coated dishes and harvested in the incubator. Next day, we transfected GST- or GJA1-20k-GFP, mCherry-DRP1, or Drp1K38A (0.5 µg/dish) with or without LifeAct-mCherry (1.0 µg/dish) plasmids as described previously (*Fu et al., 2017*), using FuGene HD (Promega, Madison, WI) following manufacturer protocol. The constructs are driven by CMV promoter and internal methionine in GJA-20k was mutated to leucine to express a single isoform as described previously (*Smyth and Shaw, 2013*). After overnight transfection, the cells were subjected to imaging or protein extraction. To obtain enough proteins from 100 mm culture dish, we multiplied the plasmid concentration based on bottom surface area of 35 mm dish (approximately 0.05 µg/cm²). To knock-down Gja1, DRP1, or DNM2, we used Gja1 siRNA (Thermo Fisher Scientific, ID HSS178257), DRP1 siRNA (Thermo Fisher Scientific, ID 19561), DNM2 siRNA (Thermo Fisher Scientific, ID s4212), and Stealth RNAiTM (Thermo Fisher Scientific) as negative control. We transfected 25 pmol of siRNA by LipofectamineTM RNAiMAX (Thermo Fisher Scientific) following manufacturer protocol. After overnight incubation, GST- or GJA1-20k-GFP plasmid was transfected as described above and the samples were subjected to subsequent experiments. To confirm the knock-down, we used six-well culture plate and proceed knock-down exactly same way and same time followed by protein extraction described below. For pharmacological DRP1 inhibition, a mitochondrial division inhibitor 1 (Mdivi-1, 50 µM) or the equal amount of Dimethyl sulfoxide (DMSO) diluted in culture medium was added to the cells at the same time as the plasmid transfection. After overnight incubation, the samples were subjected to imaging. To disrupt the actin polymerization, the cells after the transfection were incubated for 1 hour with Latrunculin A (LatA; 100 nM) or the equal amount of DMSO diluted in culture medium. After the incubation, the samples were subjected to imaging.

## Confocal live and fixed cell imaging

We followed the protocol described previously (*Fu et al., 2017*). The imaging was performed using a Nikon Eclipse Ti imaging system with a × 100/1.49 Apo TIRF objective, a spinning disk confocal unit (Yokogawa, Tokyo, Japan) with 486, 561, and 647 nm diode-pumped solid state lasers, and an ORCA-Flash 4.0 Hamamatsu camera, controlled by NIS Elements software. For live cell imaging, the cells

were imaged in culture medium using DMEM without Phenol Red (Thermo Fisher Scientific) for snapshot or time-lapse imaging. The imaging chamber was maintained 37 °C and 5 % $CO_2$. Mitochondria were labeled by incubating 37 °C for 20 minutes with Mitotracker (200 nM, Thermo Fisher Scientific) before imaging. To measure the mitochondrial membrane potential, we used TMRE (200 nM, Cayman Chemical Company, Ann Arbor, MI) incubating for 20 min at 37 °C before imaging. For fixed cell imaging, the cells were fixed by 4 % paraformaldehyde for 30 min at room temperature. After fixation, the samples were permeabilized in 0.1 % TritonX-100 in PBS for 10 min, washed with PBS 2 × 5 min, and blocked in 5 % normal goat serum (NGS) in PBS for 2 hr at room temperature. The following primary antibodies were diluted in 1 % NGS in PBS and incubated in dark moisture chamber at 4 °C overnight; anti-GFP (1:2000, Abcam), anti-TOM20 (1:1000, Abcam), ant-DRP1 (1:250, Abcam). Next day, the samples were washed with PBS 3 × 10 min and incubated with host-matched immunoglobin cross-adsorbed secondary antibodies conjugated with Alexa Fluor 488, 555, 647 (1:500, Invitorgen, Carlsbad, CA) in 1 % NGS in PBS for 1 hr at room temperature. After washing with PBS 3 × 10 min, the samples were mounted with ProLong Gold antifade regent with DAPI (Thermo Fisher Scientific). For image analysis, we used ImageJ with the mito-morphology plugin (*Dagda, 2009*) for mitochondrial morphology analysis.

## Western blot analysis

We performed Western blot as previously described (*Basheer et al., 2018*). We harvested the cells in 100 mm culture dish or six-well plate for knock-down experiment transfected plasmid as described above. The cells were lysed by RIPA buffer (containing 50 mM Tris, 150 mM NaCl, 1 mM EDTA, 1 % TritonX-100, 1 % Sodium Deoxycholate, 1 mM NaF, 0.2 mM $Na_3VO_4$, and Halt Proteinase and Phosphatase Inhibitor Cocktail (Thermo Fisher Scientific)) to extract total protein. The lysate was sonicated on ice, rotated for 1 hr at 4 °C, and centrifuged 16,000× g for 20 min at 4 °C. The supernatant was collected as the protein sample. To extract mitochondrial fraction, we used Mitochondria Isolation Kit for Cultured Cells (Thermo Fisher Scientific) following manufacturer protocol. The protein concentration was measured using DC Protein Assay (Bio-Rad, Hercules, CA). The protein samples with sample buffer (NuPAGE LDS sample buffer (NP0007) containing 100 mM DTT) were separated using NuPAGE Bis-Tris gels (4%–12%) with 3-(N-morpholino)propanesulfonic acid (MOPS) or 2-(N-morpholino) ethanesulfonic acid (MES) running buffer (Thermo Fisher Scientific) followed by transferring to polyvinylidene difluoride (PVDF) membrane (Pall Corporation, Port Washington, NY). After the transferring, the membrane was fixed by soaking in methanol, air drying, and rewetting in methanol followed by blocking with 5 % non-fat milk or bovine serum albumin (for phosphorylated protein detection) in Tris-NaCl-Tween (TNT) buffer (containing 150 mM NaCl, 50 mM Tris (pH 8.0), and 0.1 % Tween20) for 1 hour at room temperature. We used following primary antibodies diluted in TNT buffer; anti-Cx43 C-terminus (1:2000, Sigma-Aldrich), anti-DRP1 (1:500, Abcam), anti-phospho-DRP1 at S616 (1:1000, Cell Signaling Technology, Danvers, MA), anti-phospho-DRP1 at S637 (1:1000, Cell Signaling Technology), anti-MFN1 (1:1000, Cell Signaling Technology), anti-MFN2 (1:1000, Abcam), anti-TOM20 (1:500, Santa Cruz, Dallas, TX), anti-Tubulin (1:2000, Abcam, Cambridge, United Kingdom), anti-GFP (1:10000, Abcam), anti-PGC-1α (1:500, Novus Biologicals, Littleton, CO), anti-mtTFA (1:1000, Abcam), anti-LC3B (1:3000, Abcam), anti-DNM2 (1:1000, Abcam), anti-actin (1:2000, Sigma-Aldrich), anti-COX IV (1:1000, Abcam), and anti-MEK1/2 (1:1000, Cell Signaling Technology). After overnight primary antibody incubation at 4 °C, the membrane was washed with TNT buffer 3 × 10 min and incubated with host-matched immunoglobin cross-adsorbed secondary antibodies conjugated with Alexa Fluor 488, 555, 647 (1:500, Thermo Fisher Scientific) diluted in TNT buffer for 1 hour at room temperature. The membrane was washed with TNT buffer 3 × 10 min, soaked in methanol, and air dried followed by the band detection using Chemidoc MP imaging system (Bio-Rad). The band intensity was quantified using Image Lab software (Bio-Rad).

## Protein purification

GJA1-20k without the putative transmembrane region (amino acids 236 - 382 of the full-length human Cx43, NCBI reference NP 000156,1) was fused at the N-terminus with a 6 × His tag and a linker, and at the C-terminus with a linker and 10 Aspartic Acids. The fusion construct was cloned into the pET301/CT-DEST vector via Gateway cloning (Thermo Fisher Scientific). The primers to amplify the sequence are: Forward primer: 5' GGGGACAAGTTTGTACAAAAAAGCAGGCTTCAGGAGGTATACAT

ATGCATCATCATCATCATCACGGTGGTGGCGGTTCAGGCGGAGGTGGCTCTGTTAAGGATCGGG TTAAGGGAAAG 3'. Reverse primer: 5' GGGGACCACTTTGTACAAGAAAGCTGGGTCTTACTAATCG TCATCATCGTCATCATCGTCATCATCACTTCCACCACTTCCACCGATCTCCAGGTCATCAGGCCG 3'. The fusion protein was expressed in the *E. coli* expression strain LOBSTR (*Andersen et al., 2013*) (Kerafast, Boston, MA) transformed with pTf16, which encodes the chaperone protein tag (TaKaRa Bio, Shiga, Japan), following the manufacturer protocol. Protein expression was induced at 37 °C with 1 mM isopropyl-1-thio-β-galactopyranoside. The bacterial pellet containing induced GJA1-20k was lysed in B-PER Bacterial Protein Extraction Reagent (Thermo Fisher Scientific) containing cOmplete ULTRA Protease Inhibitor Tablets (Sigma-Aldrich) and sonicated and centrifuged to separate soluble proteins. The HisPur Cobalt Purification Kit (Thermo Fisher Scientific) was used to purify the protein following the manufacturer protocol with some buffer substitutions. We used a buffer containing 300 mM NaCl, 50 mM $NaH_2PO_4$, 5 mM 2-mercaptoethanol, pH 8.0 for column equilibration and washing (with addition of 10 mM and 20 mM imidazole for sequential washing steps), and we used an elution buffer containing 300 mM NaCl, 50 mM $NaH_2PO_4$, 150 mM Imidazole, pH 8.0, 10 % glycerol. Before use, the purified protein was concentrated and subjected to a buffer exchange into a final buffer containing 50 mM $NaH_2PO_4$, 150 mM NaCl (pH 8.0) with 10 % glycerol.

## Cell-free pyrene-actin polymerization and depolymerization assay

The pyrene-actin polymerization and depolymerization assays were performed using the Actin Polymerization Biochem Kit (Cytoskeleton, Inc, Denver, CO) following the manufacturer protocol. Briefly, lyophilized pyrene-conjugated skeletal muscle actin was reconstituted and diluted to a concentration of 1 mg/ml in G-Buffer (5 mM Tris-HCl pH 8.0, 0.2 mM $CaCl_2$, 0.2 mM ATP) and polymerized with 0.25× Actin Polymerization Buffer (500 mM KCl, 20 mM $MgCl_2$, 50 mM guanidine carbonate, 10 mM ATP) for 1 hr at room temperature. F-actin was incubated with either 15 nM LatA or an equal volume of DMSO, and/or 1 mM GJA1-20k protein purified as described above. For the polymerization assay, actin was incubated with 1 mM GJA1-20k protein before polymerization. The changes in fluorescence were measured in microplate reader (FlexStation 3, Molecular Devices, San Jose, CA) at excitation wavelength 365 nm and emission wavelength 407 nm. The data was obtained from duplicate or triplicate experimental repeats.

## Seahorse mitochondrial respiration assay

The mitochondrial oxygen consumption rate (OCR) was measured using Seahorse XF96 analyzer (Agilent, Santa Clara, CA) following the manufacturer protocol and previous study (*Zhang et al., 2012*). Isolated mouse neonatal cardiomyocytes (CM) described above ($8.0 \times 10^4$ cells/well) or GST- or GJA1-20k-transfected HEK293 cells ($1.0 \times 10^4$ cells/well) were seeded into Seahorse XF96 plate. Before seeding neonatal CM, the plate was coated by laminin (20 µg/ml). After 1 (HEK293) or 2 days (neonatal CM) incubation at 37 °C, OCR was measured in three time points after injection of oligomycin (2 µM for neonatal CM; 1 µM for HEK293), arbonyl cyanide 4-(trifluoromethoxy)phenylhydrazone (FCCP; 5 µM for neonatal CM; 1.5 µM for HEK293), and rotenone/antimycin A (R/A; 1 µM each for both neonatal CM and HEK293). The data was normalized by protein concentration in each well. The maximum respiration was calculated by subtracting the third measurement point of R/A treatment from that of FCCP treatment. For basal respiration, we calculated the difference between the third measurement during the pre-oligomycin phase and the third measurement post-R/A. For proton leak, we calculated the difference between the third measurement during the post-oligomycin and the third measurement post-R/A. For maximum respiration, we calculated the difference between the third measurement during the post-FCCP and the third measurement post-R/A. For ATP-linked respiration, we calculated the difference between the third measurement during the pre-oligomycin phase and the third measurement post-oligomycin. For reserve capacity, we calculated the difference between the third measurement during the post-FCCP and the third measurement the pre-oligomycin phase. For non-mitochondrial respiration, we used the value at the third measurement post-R/A.

## Mitochondrial ROS measurement

The cells were harvested in 35 mm glass-bottomed dish and transfected plasmid as described above. The samples were treated with $H_2O_2$ (600 µM, Sigma-Aldrich) or PBS as control diluted in culture medium for 30 min at 37 °C followed by MitoSOX Red Mitochondrial Superoxide Indicator (5 µM,

Thermo Fisher Scientific) for 10 min at 37 °C or CellROX Deep Red (5 μM, Thermo Fisher Scientific) for 30 min at 37 °C. After the incubation, the samples were subjected to live cell imaging. The intensity was calculated using the formula; Total cell intensity – (Area of the cell× Mean intensity of background) by Image J.

### Langendorff-perfused mouse heart I/R injury

Adult (14- to 16-week-old) male WT or Gja1$^{M213L/WT}$ mice were used for I/R study, as previously described (*Basheer et al., 2018*) with some modifications. Briefly, mice were injected with 100 U (IP) of heparin 20 min prior to IP administration of sodium pentobarbital (200 mg/kg) and hearts were removed quickly by a midsternal incision and placed into ice-cold modified pH 7.4 Krebs-Henseleit (K-H) solution. Then, the heart was attached to Langendorff apparatus, and perfused through the aorta at a constant rate 2.5 ml/min with the Krebs-Henseleit (K-H) solution (pH 7.4) gassed with 95 % $O_2$/5 % $CO_2$ constantly. The K-H solution temperature was maintained at 37 °C by circulating water bath. After 20 min equilibration to achieve a steady state, the hearts were subjected to 30 min of ischemia (no-flow ischemia) followed by 60 minutes of reperfusion. During ischemia, the hearts were immersed in warm K-H solution. After reperfusion, the hearts were sliced in 1 mm thickness and immersed in freshly made 1 % triphenyl tetrazolium chloride (TTC) solution for 20 minutes at 37 °C followed by fixation with 4 % paraformaldehyde for 30 min. Scanned image was analyzed using Image J (NIH). The area of the infarcted region of each slice was measured by semiautomatic threshold color setting, and expressed as a percentage of the total slice area. Infarct size was corrected to the weight of each slice as previously described (*Basheer et al., 2018*). For dihydroethidium (DHE; Thermo Fisher Scientific) staining, the heart sections embedded in OCT compound were made in 10 μm and wash with PBS 2 × 5 min followed by 20 min incubation with 5 μM of DHE at 37 °C. After repeat washing step with PBS, the sections were mounted by ProLong Gold Antifade Mountant with DAPI (Thermo Fisher Scientific) and subjected to fluorescent imaging.

### Mitochondrial DNA copy number analysis

Total DNA was extracted using NucleoSpin Tissue (MACHEREY-NAGEL, Düren, Germany) from the mouse hearts or the cells cultured in six-well plate following manufacturer protocol. The DNA concentration was measured using NanoDrop 2000 (Thermo Fisher Scientific) and 5 ng/μl of DNA was subjected to Real-Time PCR analysis (Bio-Rad) using Mouse Mitochondrial DNA Copy Number Assay kit (Detroit R&D, Detroit, MI) for the samples from the mouse heart or Human Mitochondrial DNA Monitoring Primer Set (TaKaRa Bio) for the cells following manufacturer protocol.

### ATP measurement

GST- or GJA1-20k-transfected HEK293 cells ($1.0 × 10^4$ cells/well) were seeded into white 96-well microplate (PerkinElmer, Waltham, MA). ATP measurement was performed using ATPlite Luminescence Assay System (PerkinElmer) following manufacturer protocol and the luminescence was measured in microplate reader (FlexStation 3, Molecular Devices). Note that the cells were washed with PBS before adding lysis buffer and 10 μl of the lysate was used for protein concentration measurement as described above. The result of ATP amount was normalized by total protein amount in each well.

### Statistical analysis

Graphs were created and analyzed using Prism six software (GraphPad). For two groups comparison, unpaired two-tail Mann-Whitney U-test was performed. For among multiple group comparison, Kruskal-Wallis test with Dunn's post-hoc test or two-way ANOVA with Bonferroni's post-hoc test was performed. p values of less than 0.05 was considered significant. All data points, exact p values, and statistical data are provided in the source data.

## Acknowledgements

We acknowledge W Basheer and D Hernandez for the technical discussion, S Ryazantsev and L Nikolova for electron microscopy, A Laxman for Seahorse analysis. JR is an Investigator of the Howard Hughes Medical Institute.

## Additional information

### Funding

| Funder | Grant reference number | Author |
|---|---|---|
| National Institutes of Health | R01HL094414 | Robin Mark Shaw |
| National Institutes of Health | R01HL138577 | Robin Mark Shaw |
| National Institutes of Health | GM131845 | Jared Rutter |
| National Institutes of Health | R01HL133286 | Tingting Hong |

The funders had no role in study design, data collection and interpretation, or the decision to submit the work for publication.

### Author contributions

Daisuke Shimura, Conceptualization, Data curation, Formal analysis, Investigation, Methodology, Project administration, Resources, Validation, Visualization, Writing – original draft, Writing – review and editing; Esther Nuebel, Rachel Baum, Data curation, Formal analysis, Investigation, Methodology, Validation, Visualization, Writing – review and editing; Steven E Valdez, Data curation, Methodology; Shaohua Xiao, Formal analysis, Investigation, Methodology, Validation, Writing – review and editing; Junco S Warren, Data curation, Formal analysis, Investigation, Methodology, Validation, Writing – review and editing; Joseph A Palatinus, Methodology, Supervision; TingTing Hong, Conceptualization, Funding acquisition, Project administration, Resources, Supervision, Validation, Writing – review and editing; Jared Rutter, Conceptualization, Formal analysis, Funding acquisition, Methodology, Project administration, Resources, Supervision, Validation, Writing – review and editing; Robin M Shaw, Conceptualization, Formal analysis, Funding acquisition, Methodology, Project administration, Resources, Supervision, Validation, Visualization, Writing – original draft, Writing – review and editing

### Author ORCIDs

Daisuke Shimura (iD) http://orcid.org/0000-0002-9954-2162
Jared Rutter (iD) http://orcid.org/0000-0002-2710-9765
Robin M Shaw (iD) http://orcid.org/0000-0001-7429-6092

### Ethics

All animal care and study protocols were approved by University of Utah Institutional Animal Care and Use Committee (#19-09010).

### Decision letter and Author response

Decision letter https://doi.org/10.7554/eLife.69207.sa1
Author response https://doi.org/10.7554/eLife.69207.sa2

## Additional files

### Supplementary files

• Transparent reporting form
• Source data 1. Uncropped membranes for all western blots in the manuscript.

### Data availability

The data supporting the findings of this study are available within the paper and its source data files. Plasmids are available at the non-profit plasmid repository Addgene, under Robin Shaw Lab.

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

## Appendix 1

**Appendix 1—key resources table**

| Reagent type (species) or resource | Designation | Source or reference | Identifiers | Additional information |
|---|---|---|---|---|
| Strain, strain background (*Mus musculus*) | C57BL/6 J | The Jackson Laboratory | #000664 | |
| Strain, strain background (*Mus musculus*) | Gja1 M213L/M213L | *Xiao et al., 2020* (DOI: 10.1172/JCI134682) | | |
| Strain, strain background (*AAV9*) | AAV9-GST-GFP | Welgen Inc *Basheer et al., 2018* (DOI: 10.1172/jci. insight.121900) | | |
| Strain, strain background (*AAV9*) | AAV9-GJA1-20k-GFP | Welgen Inc *Basheer et al., 2018* (DOI: 10.1172/jci. insight.121900) | | |
| Strain, strain background (*Aenovirus*) | GJA1-20k-V5 | the CURE Vector Core Facility at University California, Los Angeles *Basheer et al., 2017* (DOI: 10.1161/ CIRCRESAHA.117.311955) | | |
| Strain, strain background (*Aenovirus*) | GFP-V5 | the CURE Vector Core Facility at University California, Los Angeles *Basheer et al., 2017* (DOI: 10.1161/ CIRCRESAHA.117.311955) | | |
| Strain, strain background (*E. coli*) | LOBSTR *E. coli* Expression Strain | kerafast *Andersen et al., 2013*; *Figure 4—figure supplement 1—source data 1* (DOI: 10.1002/prot.24364) | EC1001 | |
| Cell line (*Homo-sapiens*) | HEK293FT | Thermo Fisher Scientific | R70007 | |
| Transfected construct (human) | siRNA to Gja1 | Thermo Fisher Scientific | ID HSS178257 | |
| Transfected construct (human) | siRNA to DRP1 | Thermo Fisher Scientific | ID 19561 | |
| Transfected construct (human) | siRNA to Dynamin 2 | Thermo Fisher Scientific | ID s4212 | |
| Antibody | anti-Cx43-CT (Rabbit polyclonal) | Sigma-Aldrich | C6219 | WB (1:2000) |
| Antibody | anti-DRP1 (Mouse monoclonal) | Abcam | Ab56788 | WB (1:500) ICC (1:250) |
| Antibody | anti-phospho-DRP1 at S616 (Rabbit monoclonal) | Cell Signaling Technology | 4,494 S | WB (1:1000) |
| Antibody | anti-phospho-DRP1 at S616 (Rabbit polyclonal) | Cell Signaling Technology | 4,867 S | WB (1:1000) |
| Antibody | anti-MFN1 (Rabbit monoclonal) | Cell Signaling Technology | 14,739 S | WB (1:1000) |

*Appendix 1 Continued on next page*

*Appendix 1 Continued*

| Reagent type (species) or resource | Designation | Source or reference | Identifiers | Additional information |
|---|---|---|---|---|
| Antibody | anti-MFN2 (Mouse monoclonal) | Abcam | ab56889 | WB (1:1000) |
| Antibody | anti-MFN2 (Mouse monoclonal) | Abcam | ab56889 | WB (1:1000) |
| Antibody | anti-TOM20 (Mouse monoclonal) | Santa Cruz | sc-17764 | WB (1:500) |
| Antibody | anti-TOM20 (Rabbit polyclonal) | Abcam | ab78547 | ICC (1:1000) |
| Antibody | anti-Tubulin (Rat monoclonal) | Abcam | ab6160 | WB (1:2000) |
| Antibody | anti-GFP (Chicken polyclonal) | Abcam | ab13970 | WB (1:10000) ICC (1:2000) |
| Antibody | anti-actin (Rabbit polyclonal) | Sigma-Aldrich | A2103 | WB (1:2000) |
| Antibody | anti-COX IV (Mouse monoclonal) | Abcam | ab14744 | WB (1:1000) |
| Antibody | anti-MEK1/2 (Mouse monoclonal) | Cell Signaling Technology | 4,694 S | WB (1:1000) |
| Recombinant DNA reagent | pDEST-GJA1-20k-GFP (plasmid) | *Fu et al., 2017* (DOI: 10.3389/fphys.2017.00905) | | GFP version of Addgene_#49,861 |
| Recombinant DNA reagent | pDEST-GST-GFP (plasmid) | *Fu et al., 2017* (DOI: 10.3389/fphys.2017.00905) | | |
| Recombinant DNA reagent | pDEST-LifeAct-mCherry (plasmid) | Addgene | 40,908 | |
| Recombinant DNA reagent | pDEST-LifeAct-mCherry (plasmid) | Addgene | 40,908 | |
| Recombinant DNA reagent | mCherry-Drp1 (plasmid) | Addgene | 49,152 | |
| Recombinant DNA reagent | pcDNA3-Drp1K38A (plasmid) | Addgene | 45,161 | |
| Sequenced-based reagent | siRNA: Negative Control | Thermo Fisher Scientific | 12935300 | Stealth RNAi |
| Sequenced-based reagent | Gja1_Fw | This paper | PCR primer | GGGGACAAGTTTGTAC AAAAAAGCAGGCTT CAGGAGGTATACATAT GCATCATCATCATCAT CACGGTGGTGGCGGTT CAGGCGGAGGTGG CTCTGTTAAGGATCGG GTTAAGGGAAAG |
| Sequenced-based reagent | Gja1_Rv | This paper | PCR primer | GGGGACCACTTTGTAC AAGAAAGCTGGGTC TTACTAATCGTCATCA TCGTCATCATCGTCATC ATCACTTCCACCACTT CCACCGATCT CCAGGTCATCAGGCCG |
| Peptide, recombinant protein | GJA1-20k | This paper | | amino acids 236–382 of the full-length human Cx43, NCBI reference NP 000156,1 |

*Appendix 1 Continued on next page*

*Appendix 1 Continued*

| Reagent type (species) or resource | Designation | Source or reference | Identifiers | Additional information |
|---|---|---|---|---|
| Commercial assay or kit | DC Protein Assay | Bio-Rad | 5000116 | |
| Commercial assay or kit | Actin Polymerization Biochem Kit | Cytoskeleton, Inc | BK003 | |
| Commercial assay or kit | Mouse Mitochondrial DNA Copy Number Assay kit | Detroit R&D | NC1134958 | |
| Commercial assay or kit | Human Mitochondrial DNA Monitoring Primer Set | TaKaRa Bio | 7,246 | |
| Commercial assay or kit | Primary Cardiomyocyte Isolation Kit | Life Technologies | 88,281 | |
| Commercial assay or kit | Mitochondria Isolation Kit | Thermo Fisher Scientific | 89,874 | |
| Commercial assay or kit | ATPlite Luminescence Assay System | PerkinElmer | 6016943 | |
| Commercial assay or kit | HisPur Cobalt Purification Kit | Thermo Fisher Scientific | PI-90092 | |
| Chemical compound, drug | FuGene HD | Promega | E2312 | |
| Chemical compound, drug | Lipofectamine RNAiMAX | Thermo Fisher Scientific | 13778150 | |
| Chemical compound, drug | mitochondrial division inhibitor 1 | Sigma-Aldrich | M0199 | 50 µM |
| Chemical compound, drug | Latrunculin A | Sigma-Aldrich | L5163 | 100 nM |
| Chemical compound, drug | Mitotracker Deep Red | Thermo Fisher Scientific | m22426 | 200 nM |
| Chemical compound, drug | Mitotracker Red CMXRos | Thermo Fisher Scientific | M7512 | 200 nM |
| Chemical compound, drug | TMRE | Cayman Chemical Company | 701,310 | 200 nM |
| Chemical compound, drug | MitoSOX Red Mitochondrial Superoxide Indicator | Thermo Fisher Scientific | M36008 | 5 µM |
| Chemical compound, drug | CellROX Deep Red | Thermo Fisher Scientific | C10422 | 5 µM |
| Chemical compound, drug | ProLong Gold antifade regent with DAPI | Thermo Fisher Scientific | P36935 | |
| Software, algorithm | GraphPad Prism 6.0 | GraphPad Software Inc | Windows | |
| Software, algorithm | ImageJ | NIH | Windows | |
| Software, algorithm | Mitochondria morphology plugin | Dagda, Cherra et al. 2009 (DOI: 10.1074/jbc. M808515200) | | |

*Appendix 1 Continued on next page*

*Appendix 1 Continued*

| Reagent type (species) or resource | Designation | Source or reference | Identifiers | Additional information |
|---|---|---|---|---|
| Software, algorithm | Chemidoc MP imaging system | Bio-Rad | | |
| Software, algorithm | Image Lab software | Bio-Rad | | |
| Software, algorithm | Adobe Photoshop | Adobe | Version 22.4.2 | |
| Software, algorithm | Adobe Illustrator | Adobe | Version 25.3.1 | |
| Other | triphenyl tetrazolium chloride (Stain) | Sigma-Aldrich | T8877 | |
| Other | dihydroethidium (Stain) | Thermo Fisher Scientific | D23107 | |

