## [Decision Letter]

**Acceptance summary:**

This study identifies a cardioprotective factor, GJA1-20k (a truncated form of Cx43), which appears to confer protection against ischemia/reperfusion (I/R) injury via promotion of mitochondrial fission. This finding is particularly interesting given that hyperfission is generally thought of as an index of toxicity in I/R or hypoxic injury. I/R lesion size in a GJA1 heterozygous mutant mouse is strikingly exacerbated compared to control animals, providing strong in vivo evidence supporting a role for this factor in protection from I/R.

**Decision letter after peer review:**

Thank you for submitting your article "Protective mitochondrial fission induced by stress responsive protein GJA1-20k" for consideration by *eLife*. Your article has been reviewed by 3 peer reviewers, and the evaluation has been overseen by a Reviewing Editor and Matt Kaeberlein as the Senior Editor. The following individual involved in review of your submission has agreed to reveal their identity: Jason A Mears (Reviewer #3).

Essential Revisions

Mdivi-1 is has been shown to inhibit ETC CI as well as Drp1 (see "The Putative Drp1 Inhibitor mdivi-1 Is a Reversible Mitochondrial Complex I Inhibitor that Modulates Reactive Oxygen Species"). Changes to mitochondrial dynamics/structure may result from effects on ETC CI. Given the importance of Drp1 independence in the conclusions, the authors need to validate their findings using another method, such as overexpression of the dominant negative Drp1 mutant K38A.

K38A (or a similar approach) should also be used for the kinetic experiments.

The relative change in GJA1-20k expression in heterozygous (M213L/WT) animals should be shown, and the relative changes to ETC function (Seahorse) and mitochondrial size should be shown for the heterozygous animals. Some of this is already in supplemental, it should be presented in the main figures. This is important for interpreting the data in figure 5. The specificity of the phenotype to the stressed condition is striking, but because the phenotype is only shown in the heterozygotes (as homozygosity is lethal) the descriptive data in prior figures needs to show this setting.

The authors should show mitochondrial size post-I/R in control and heterozygous animals.

The H2O2 induced oxidative stress assay is not particularly informative or easy to interpret. I recommend moving this data to the supplement and focusing on additional endpoints from the mouse model in figure 5. The robustness of the phenotype in the I/R assay is much more compelling.

Mechanism of Drp1 independent fission is unclear – due to other GTPases such as dynamin-2?

The role of increased fission in protecting against ROS production thought to be a particularly interesting finding, but reviewers believe that mitochondria fragmented by Drp1 overexpression and suppression of Mfn1/2 or Opa1 is needed as an additional control.

To begin to address whether the fission/ROS findings from HEK293 cells are relevant to the in vivo results, the authors should be repeated in a more relevant cell type, such as in neonatal cardiomyocyte cells.

The authors should address the impact of GJA1-20k on mitochondrial membrane potential, ATP generating capacity, and rates of mitophagy.

GJA1-20k induces a clear difference in mitochondrial size, while total mitochondrial content appears unaltered (by Tom20). Previously it was suggested that mitochondrial biogenesis occurs with increased levels of GJA1-20k. Is this a difference in the cellular model, and do the changes in cell culture accurately recapitulate the changes seen in animals?

Please comment on the role of GJA1-20k in pathologic mitochondrial division during hypoxia/reoxygenation.

Please discuss the any perceived therapeutic implications of the findings.

Please discuss how altered mitochondrial size may alter cardiomyocyte and cardiac function. Is it possible that the benefits of GJA1-20K to infarct size come at a cost of cardiac function and reserve? Additional data to address this possibility would be helpful, but at minimum a discussion is needed.

*Reviewer #1 (Recommendations for the authors):*

This group originally found that cardiomyocyte-specific overexpression of the Cx43 truncated isoform (GJA1-20k) protects the heart against ischemia/reperfusion injury in mice, and this study support their previous study by demonstrating the cardioprotective effect of GJA1-20k peptide administration. The authors suggest that Drp1-independent mitochondrial fission and ROS production precondition the heart against severe ischemic stress. This concept is very interesting and novel, but there are several concerns that should be revised.

1. The mechanism of how GJA1-20k induces Drp1-independent fission via actin accumulation to mitochondria is unclear. Please identify what drives the mitochondrial fission. Is it due to other GTPases such as dynamin-2 Nature 540, 139-143 (2016). https://doi.org/10.1038/nature20555?

2. The most important point is that GJA1-20k causes mitochondrial fission with few ROS production, and attenuates mitochondrial ROS production caused by hydrogen peroxide. To avoid the possibility of an artifact imaging, they need to carry out comparative experiments with cells that have fragmented mitochondria due to overexpression of Drp1 and suppression of Mfn1/2 or Opa1.

3. Most cell-level experiments use HEK293 cells, and it remains questionable whether they reflect in vivo cardiac results. Since the adenoviral gene transfer system into neonatal cardiomyocytes is performed in Figure 1, it is better to see the production of mitochondrial ROS in the same system (Figure 5E).

4. Does the GJA1-20k overexpression have impact on mitochondrial membrane potential and ATP-producing ability? Is mitochondria that are divided by GJA1-20k removed through mitophagy?

5. I would like you to clarify how much this GJA1-20k-dependent mitochondrial division contributes to pathological mitochondrial division under hypoxia /reoxygenation, which can be separated from Drp1-dependent mitochondrial fission. Figure 5—figure supplement 1 compares the mitochondria of WT and Gja1M213L / WT in the physiological state.

*Reviewer #2 (Recommendations for the authors):*

The following manuscript from Shimura et al., seeks to investigate the role of a 20kD protein isoform of Cx43 (GJA1-20k) in ischemic preconditioning. The authors discovered that GJA1-20K may provide protection in ischemic hearts through polymerizing actin around mitochondria and inducing mitochondrial fission. Through a series of elegant genetic, chemical, biochemical and cell biology studies including the use of the Gja1 M213L mouse line which was generated by this group, the authors determined that the salutatory effects of GJA1-20K was due to decreased reactive oxygen species (ROS) generation from smaller mitochondria. In general, the studies are a nice extension of published work from this group who initially discovered that a wide range of Cx43 protein isoforms can be generated from internal translation initiation sites within this protein. In particular, this study now further explores the specific role of GJA1-20K in the heart and discovered that it plays a critical role in ischemic preconditioning through affecting mitochondrial size and ROS production. Although the work is well done, there are some outstanding issues that the authors should address. It would be helpful if the authors discuss more the implication of the GJA1-20K isoform as therapy as the AAV9 work suggests the possibility of gene therapy in the future. It remains unclear as to how the authors envision such a scenario. Additional discussion would be helpful as to how smaller mitochondria from GJA1-20K over expression might impact cardiomyocyte and overall cardiac function, especially after ischemia or infarction. Although GJA1-20K may reduce infarct size, could this be at the expense of overall cardiac function or reserve?

*Reviewer #3 (Recommendations for the authors):*

1. What are the levels of GJA1-20k over-expression in expts? Is it comparable to what has been described for endogenous increase observed with cardiomyocytes under stress?

2. Is there any difference when comparing mitochondria in the SSM and IFM? Are both populations affected?

3. The 20 kDa peptide contains some TM sequence, so why is it not targeted to the PM? Is there a MTS? A schematic earlier in the text would help orient the reader – similar to what has been shown in previous articles.

4. No change in S616 phosphorylation, but why not check S637? This site is thought to be the major driver of Drp1 activation/inactivation. So changes in the phosphorylation status at 637 is critical to provide a complete evaluation of Drp1 PTM regulation.

5. Is Drp1 localization altered when GJA1-20k is overexpressed? Hard to tell from Supplemental Figures No quantification.

6. In Figure 5, the Seahorse data show a higher level of non-mitochondrial respiration in the animal samples. This difference also mutes the differences in Basal Respiration, Leak and Maximal Respiration. There is definitely a trend in the data, but the differences are not as convincing if the non-mitochondrial respiration is taken into account (unless I am missing something in the way that the data is presented).

7. IMO, the discussion is a little long and winding. Some of it is necessary, but I think it can be more efficiently organized to better highlight the impact of your findings.

8. Typo in Line 45… "stress sFriedman and".

---

## [Author Response]

Reviewer #1 (Recommendations for the authors):This group originally found that cardiomyocyte-specific overexpression of the Cx43 truncated isoform (GJA1-20k) protects the heart against ischemia/reperfusion injury in mice, and this study support their previous study by demonstrating the cardioprotective effect of GJA1-20k peptide administration. The authors suggest that Drp1-independent mitochondrial fission and ROS production precondition the heart against severe ischemic stress. This concept is very interesting and novel, but there are several concerns that should be revised.1. The mechanism of how GJA1-20k induces Drp1-independent fission via actin accumulation to mitochondria is unclear. Please identify what drives the mitochondrial fission. Is it due to other GTPases such as dynamin-2 Nature 540, 139-143 (2016). https://doi.org/10.1038/nature20555?

We thank the Reviewer for raising this point and appreciate the novelty of introducing a type of mitochondrial fission that is independent of known mediators of fission. To address the Reviewer’s question and explore whether dynamin-2 (DNM2) is involved in the mitochondrial fission with GJA1-20k, for the revised manuscript we knocked-down DNM2 and assessed mitochondrial morphology with or without GJA1-20k. As seen in revised Figure 2—figure supplement 1 panels G–I, DNM2 knock-down does not significantly affect overall average area of mitochondria, with or without the presence of GJA1-20k. The failure of DNM2 knockdown to affect GJA1-20k mediated fission in the HEK cells we used, suggests that GJA1-20k activity on mitochondria can be independent of DNM2. Clearly there is still much to learn about GJA1-20k mediated fission. GJA1-20k is associated with the mitochondrial outer membrane (Basheer *et al.*, *JCI Insight*, 2018). While we should not rule out other mediators, we expect GJA1-20k is a direct binding partner and regulator of actin dynamics, as suggested by Figures 3D and 3E, and is therefore responsible for actin organization around mitochondrial bodies, inducing an actin mediated scission event (Figure 4).

The revised manuscript has been modified to include the above new data (Figure 2—figure supplement 1G–I) and discussion:

— Results section

“In addition to DRP1, Dynamin-2 (DNM2) is also a GTPase which can potentially regulate mitochondrial fission (Lee, Westrate et al., 2016). To explore possible DNM2 involvement, we knocked-down DNM2 by siRNA and analyzed mitochondrial morphology. In contrast to siDRP1, siDNM2 alone did not affect mitochondrial morphology (Figure 2—figure supplement 1G – I), with or without GJA1-20k, suggesting no involvement of DNM2 in mitochondrial dynamics of HEK293 cell lines. It has been reported that DNM2 is not necessary to be responsible for mitochondrial fission in other cell lines as well (Fonseca, Sanchez-Guerrero et al., 2019). The fact that DNM2 is not needed for GJA1-20k activity to occur, indicates it is not an essential co-factor in GJA1-20k mediated mitochondrial fission.”

—Discussion sections

“We have also investigated the involvement of DNM2 as another GTPase for the mitochondrial fission, but DNM2 knock-down failed to induce sufficient mitochondrial fusion or alter the effects of GJA1-20k (Figure 2 —figure supplement 1G–I). Therefore, it does not appear that DNM2 is essential for GJA1-20k activity on mitochondrial fission. In addition to actin, the endoplasmic reticulum (ER) membrane can be involved in mitochondrial scission (Friedman, Lackner et al., 2011, Tandler, Hoppel et al., 2018). Future studies should be considered whether GJA1-20k induced actin cytoskeleton arrangements involves ER membrane as well. Given the rapidity (less than one minute) of GJA1-20k induced fission, it is of interest how fission activity can be limited in stable cells, especially terminally differentiated cells such as adult cardiomyocytes.”

2. The most important point is that GJA1-20k causes mitochondrial fission with few ROS production, and attenuates mitochondrial ROS production caused by hydrogen peroxide. To avoid the possibility of an artifact imaging, they need to carry out comparative experiments with cells that have fragmented mitochondria due to overexpression of Drp1 and suppression of Mfn1/2 or Opa1.

We appreciate the Reviewer’s excellent suggestion. To address this concern, we overexpressed DRP1 and measured ROS production under H2O2 stress. As expected (and predicted by the Reviewer), DRP1 overexpression caused fragmented mitochondria (new Figure 5—figure supplement 1F, G). In addition, while H2O2 did not induce an increase in ROS production in the presence of GJA1-20k (new Figure 5—figure supplement 1D, E), H2O2 did induce an increase in ROS in the presence of DRP1 overexpression (new Figure 5—figure supplement 1 H, I). Therefore, mitochondrial fragmentation caused by DRP1 overexpression is more sensitive to oxidative stress and produces higher amount of ROS than in the presence of GJA1-20k. These data are supportive to our conclusion that GJA1-20k induces protective (rather than harmful) mitochondrial fission.

In addition, to add to the rigor of the study and address concerns about cardiac specificity, we performed new intact heart experiments revealing that there are larger mitochondria, more ROS, and larger areas of infarction in GJA1-20k insufficient mice subjected to I/R injury (Figure 5 E–L in the revised manuscript).

The revised manuscript has been modified to include the above new data (Figure 5E–L, and Figure 5—figure supplement 1) and discussion:

—Results section

“Given that GJA1-20k is a stress-responsive protein which can mediate ischemic preconditioning (Basheer, Fu et al., 2018) and to explore the protective effects of GJA1-20k against oxidative stress, we measured mitochondrial generation of reactive oxygen species (ROS) in HEK293 cells under oxidative stress induced by exogenous H2O2 treatment. In unstressed cells, the presence of GJA1-20k does not affect mitochondrial ROS generation (Figure 5—figure supplement 1D, E) whereas in the presence of exogenous H2O2, GJA1-20k prevents an increase in mitochondrial ROS (Figure 5—figure supplement 1D, E). Thus, GJA1-20k, which depresses mitochondrial respiration in unstressed cells, does not affect homeostatic ROS generation, yet has the beneficial effect of decreasing toxic ROS generation during stress (Figure 5A, B; Figure 5—figure supplement 1D, E). To clarify whether the protective effects come from GJA1-20k expression or mitochondrial size, we measured ROS from HEK293 cells with DRP1 overexpression instead of GJA1-20k overexpression. DRP1 overexpression decreased in mitochondrial size similar to GJA1-20k (Figure 5—figure supplement 1F, G). However, ROS generation induced by H2O2 was significantly increased and higher than GST control under H2O2 (Figure 5—figure supplement 1H, I). Taken together, GJA1-20k itself is necessary for the protective mitochondrial fission.”

—Discussion section

“DRP1-induced fission and smaller mitochondria are associated with excess ROS production, apoptosis, and organ injury (Hu, Huang et al., 2017, Wang, Subramanian et al., 2017), which is also seen in Figure 5 —figure supplement 1F–I, whereas DRP1-independent mitochondrial fission has been observed in cells undergoing protective mitophagy (Stavru, Palmer et al., 2013, Yamashita, Jin et al,. 2016, Coronado, Fajardo et al., 2018).”

—Discussion section

“Because GJA1-20k-induced fission is associated with less ROS production with oxidative stress (Figure 5 —figure supplement 1D, E), the endogenous generation of GJA1-20k and subsequent decreased ROS production could explain a major benefit of pre-conditioning. Of note, genetic GJA1-20k reduction increases infarct size and ROS production post-I/R injury (Figure 5E–H). In addition, the population of damaged mitochondria is significantly increased in heterozygous Gja1^M213L/WT^ mouse heart post-I/R (Figure 5I–L). Therefore, GJA1-20k induced decreases in ROS production could limit the amount of I/R injury induced by myocardial infarction.”

3. Most cell-level experiments use HEK293 cells, and it remains questionable whether they reflect in vivo cardiac results. Since the adenoviral gene transfer system into neonatal cardiomyocytes is performed in Figure 1, it is better to see the production of mitochondrial ROS in the same system (Figure 5E).

This is an excellent comment from the Reviewer although we should emphasize, in the original Figure 1, we identified a consistent phenomenon of GJA1-20k induced mitochondrial reduction or GJA1-20k removal induced mitochondrial size increase in not just HEK293 cells, but neonatal cardiomyocytes, cardiomyocytes from the hearts of young adult mice as well as from the hearts of adult mice. With regard to mitochondrial ROS generation, isolated neonatal cardiomyocytes are notoriously difficult to induce mitochondrial ROS (Kang *et al.*, *Circ Res.*, 2000). Therefore, to address the Reviewer’s concern about physiological relevance, we used adult mouse heart tissue post-I/R injury and measured ROS production. As seen in new Figure 5 Panels G and H, adult heterozygous (Gja1^M213L/WT^) mouse hearts experience an increase in ROS production post-I/R injury compared to hearts from WT animals. Moreover, our previous study (Basheer *et al.*, *JCI insight*, 2018, Figure 7E, F) AAV9-transduced GJA1-20k overexpressed adult cardiomyocytes decreased ROS production. Taken together, the ROS measurements using HEK293 cells in the present manuscript are consistent with the new ex vivo adult heart results.

The revised manuscript has been modified to include the above new data (Figure 5G, H) and discussion:

—Results section

“However, remarkably, reduced GJA1-20k expression results in an almost complete cardiac infarction after I/R injury (Figure 5E, F). Moreover, ROS production after I/R injury was increased in Gja1^M213L/WT^ mice compared to WT post-I/R (Figure 5G, H). There was no significant difference in mitochondria size at the basal condition between WT and Gja1^M213L/WT^ mice adult CMs as with neonatal CMs (Figure 5I, J), whereas the mitochondria size was significantly increased after I/R injury and the heterozygous Gja1^M213L/WT^ mice had larger mitochondria compared to WT mice post-I/R (Figure 5I, J). Interestingly, the area of mitochondrial matrix was also increased, suggesting loss of cristae in Gja1^M213L/WT^ mice heart (Figure 5K, L). These data indicate that even partial deletion of GJA1-20k results in a profoundly impaired response to ischemic stress.”

—Discussion section

“Because GJA1-20k-induced fission is associated with less ROS production with oxidative stress (Figure 5 —figure supplement 1D, E), the endogenous generation of GJA1-20k and subsequent decreased ROS production could explain a major benefit of pre-conditioning. Of note, genetic GJA1-20k reduction increases infarct size and ROS production post-I/R injury (Figure 5E–H). In addition, the population of damaged mitochondria is significantly increased in heterozygous Gja1^M213L/WT^ mouse heart post-I/R (Figure 5I–L). Therefore, GJA1-20k induced decreases in ROS production could limit the amount of I/R injury induced by myocardial infarction.”

4. Does the GJA1-20k overexpression have impact on mitochondrial membrane potential and ATP-producing ability? Is mitochondria that are divided by GJA1-20k removed through mitophagy?

We appreciate the Reviewer’s excellent suggestions. To address these concerns, we analyzed mitochondrial membrane potential (measured by TMRE intensity), ATP production, and mitophagy (measured by LC3 intensity). As seen new Figure 1—figure supplement 2, GJA1-20k expression, or its homozygous removal, does not alter membrane potential, ATP production, or mitophagy.

The revised manuscript has been modified to include the above new data (Figure 1—figure supplement 2) and discussion:

—Results section

“Previously we reported that GJA1-20k is involved in mitochondrial biogenesis (Basheer, Fu et al., 2018). Consistent with our previous study, AAV9-transduced adult cardiomyocytes showed increased mitochondrial DNA copy number and GJA1-20k deficient mice (Gja1M213L/M213L) had decreased copy number. However, exogenous GJA1-20k did not alter the mitochondrial biogenesis in HEK293 cells. Nor did exogenous GJA1-20k affect membrane potential or baseline ATP production (Figure 1—figure supplement 2A–C). In addition to mitochondrial DNA copy number, neither biogenesis nor mitophagy protein markers were altered in either GJA1-20k transfected HEK293 cells or Gja1^M213L/M213L^ mouse hearts (Figure 1—figure supplement 2D – G).

—Discussion section

“Yet the presence of GJA1-20k, while inducing mitochondrial fission and smaller mitochondria (Figure 1, 3 and 4), does not either reduce MFN1 or MFN2, activate DRP1, change membrane potential, ATP production, mitochondrial biogenesis, or mitophagy (Figure 2; Figure 1 —figure supplement 2).”

5. I would like you to clarify how much this GJA1-20k-dependent mitochondrial division contributes to pathological mitochondrial division under hypoxia /reoxygenation, which can be separated from Drp1-dependent mitochondrial fission. Figure 5—figure supplement 1 compares the mitochondria of WT and Gja1M213L / WT in the physiological state.

We appreciate the Reviewer’s excellent suggestion. In the same protocol for response to point #3 above, we subjected adult WT and Gja1^M213L/WT^ hearts to ex-vivo I/R injury and then used electron microscopy to assess mitochondrial differences. Consistent with previous studies (Allen *et al.*, *Commun Biol*, 2020), acute I/R injury caused mitochondrial swelling in both WT and Gja^M213L/WT^ hearts. However, Gja1^M213L/WT^ hearts had a greater increase in mitochondrial swelling compared to WT post-I/R (Figure 5I, J). Furthermore, Gja1^M213L/WT^ hearts had increased mitochondria matrix area (i.e. less cristae), indicating additional fragility of mitochondria from Gja1^M231L/WT^ hearts (Figure 5I, K, and L). These data are supportive of the previous TTC staining results (Figure 5E, F), a necessity of GJA1-20k for cardioprotection against acute I/R injury.

The revised manuscript has been modified to include the above new data (Figure 5I–L) and discussion:

—Results section

“There was no significant difference in mitochondria size at the basal condition between WT and Gja1^M213L/WT^ mice adult CMs as with neonatal CMs (Figure 5I, J), whereas the mitochondria size was significantly increased after I/R injury and the heterozygous Gja1^M213L/WT^ mice had larger mitochondria compared to WT mice post-I/R (Figure 5I, J). Interestingly, the area of mitochondrial matrix was also increased, suggesting loss of cristae in Gja1^M213L/WT^ mice heart (Figure 5K, L). These data indicate that even partial deletion of GJA1-20k results in a profoundly impaired response to ischemic stress.”

—Discussion section

“Because GJA1-20k-induced fission is associated with less ROS production with oxidative stress (Figure 5 —figure supplement 1D, E), the endogenous generation of GJA1-20k and subsequent decreased ROS production could explain a major benefit of pre-conditioning. Of note, genetic GJA1-20k reduction increases infarct size and ROS production post-I/R injury (Figure 5E–H). In addition, the population of damaged mitochondria is significantly increased in heterozygous Gja1^M213L/WT^ mouse heart post-I/R (Figure 5I–L). Therefore, GJA1-20k induced decreases in ROS production could limit the amount of I/R injury induced by myocardial infarction.”

Reviewer #2 (Recommendations for the authors):The following manuscript from Shimura et al., seeks to investigate the role of a 20kD protein isoform of Cx43 (GJA1-20k) in ischemic preconditioning. The authors discovered that GJA1-20K may provide protection in ischemic hearts through polymerizing actin around mitochondria and inducing mitochondrial fission. Through a series of elegant genetic, chemical, biochemical and cell biology studies including the use of the Gja1 M213L mouse line which was generated by this group, the authors determined that the salutatory effects of GJA1-20K was due to decreased reactive oxygen species (ROS) generation from smaller mitochondria. In general, the studies are a nice extension of published work from this group who initially discovered that a wide range of Cx43 protein isoforms can be generated from internal translation initiation sites within this protein. In particular, this study now further explores the specific role of GJA1-20K in the heart and discovered that it plays a critical role in ischemic preconditioning through affecting mitochondrial size and ROS production. Although the work is well done, there are some outstanding issues that the authors should address. It would be helpful if the authors discuss more the implication of the GJA1-20K isoform as therapy as the AAV9 work suggests the possibility of gene therapy in the future. It remains unclear as to how the authors envision such a scenario.

We agree with the Reviewer that AAV9 mediated cardioprotection of anticipated ischemia would be the natural extension of our work. In addition, gene delivery to other ischemia sensitive organs such as the kidneys and brain should be considered. There is also the possibility of delivery of GJA1-20k peptide acutely. In unpublished preliminary studies, we have found that GJA1-20k peptide in culture is able to cross the cell membrane, enter cells and increase the cells’ resistance to oxidative and fluorescent damage. Furthermore, in a large animal (porcine) model of hypovolemic shock, we have found that intravenous injection of GJA1-20k protects against experimentally induced cardiac ischemia. The revised manuscript has been modified to include this discussion:

—Discussion section

“Major cardiovascular procedures are associated with low flow and ischemia related damage to end-organs such as the heart, kidneys, and brain. A priori gene therapy mediated delivery of GJA1-20k to these organs could be beneficial. For instance, percutaneous coronary interventions are associated with both cardiac and renal damage (Brown, Malenka et al., 2008, Tricoci, Newby et al., 2018) and could benefit from GJA1-20k induced protection. Similarly, coronary bypass surgery with cardiopulmonary bypass is associated with low flow damage to the brain and kidneys (Roach, Kanchuger et al., 1996, McKhann, Grega et al., 2006, O'Neal, Shaw et al., 2016), and abdominal aorta surgery is associated with renal damage, all of which may benefit from viral mediated GJA1-20k pretreatment.

It is also possible, as suggested by unpublished preliminary studies by our group, that GJA1-20k peptide administration intravenously can gain access to the intracellular environments of the heart and other organs. In this scenario, GJA1-20k administration could occur minutes to hours ahead of an ischemic insult rather than the presumably days to weeks which would be needed for viral mediated introduction. Future studies will focus on acute intravenous administration of GJA1-20k peptide in subjects about to undergo anticipated ischemia.”

Additional discussion would be helpful as to how smaller mitochondria from GJA1-20K over expression might impact cardiomyocyte and overall cardiac function, especially after ischemia or infarction. Although GJA1-20K may reduce infarct size, could this be at the expense of overall cardiac function or reserve?

Neither viral mediated overexpression of GJA1-20k or development of a mouse line that reduces internal translation initiation of GJA1-20k results in a significant change in cardiac function (Basheer *et al.*, *JCI insight*, 2018; Xiao *et al.*, *J Clin Invest*, 2020). However, as the Reviewer suggests, acute administration of higher dose of GJA1-20k could still possibly result in decreased function. In fact, as we have found earlier (Basheer *et al.*, *JCI insight*, 2018), GJA1-20k is stress responsive and may mediate post-ischemic cardiac stunning which is typified by viable but poorly functioning myocardium post-ischemic insult. The revised manuscript has been modified to include this discussion:

—Discussion section

“We have found that neither exogenous GJA1-20k overexpression in mouse hearts nor GJA1-20kM213L/WT adult mice have altered unstressed cardiac function (Basheer, Fu et al., 2018, Xiao, Shimura et al., 2020). However acute administration of GJA1-20k or using GJA1-20k in the context of stressed myocardium could conceivably result in diminished cardiac function. As GJA1-20k is stress responsive, an increase in GJA1-20k may be a mediator of post-ischemic protection (Basheer, Fu et al., 2018), and even responsible for the phenomenon of myocardial stunning which happens to be associated with small mitochondria (Borgers, Thone et al., 1993, Borgers 2002). On the other hand, GJA1-20k induced functional hibernation of stressed myocardium, while increasing survival, could conceivably result in diminished cardiac function. If indeed GJA1-20k induces an acute cardiac depressant response, appropriate steps would be needed to compensate for lower cardiac output during the myocardial protection period.”

Reviewer #3 (Recommendations for the authors):1. What are the levels of GJA1-20k over-expression in expts? Is it comparable to what has been described for endogenous increase observed with cardiomyocytes under stress?

We appreciate Reviewer #3’s valuable question. In the present study, the exogenous plasmid transfection expresses approximately 5.6-fold increased GJA1-20k compared to endogenous (See Author response image 1 ). Our previous study revealed about 1.5-fold, 6.0-fold, and 2.0-fold increased GJA1-20k expression in mouse I/R injury, mouse subacute ischemic cardiomyopathy, and human end-stage ischemic cardiomyopathy, respectively (Figure 1 and 2 in Basheer *et al.*, *JCI insight*, 2018). Therefore, our cell line data is applicable for the GJA1-20k overexpression study, especially for acute effects.

**Author response image 1. sa2fig1:** The quantification of exogenous GJA1-20k expression in HEK293 cells. (A) Representative Western blot membrane image from GFP-tagged GJA1-20k transfected cells. (B and C) The quantification of band intensity between endogenous and exogenous GJA1-20k. The graphs were expressed as mean ± SD.

2. Is there any difference when comparing mitochondria in the SSM and IFM? Are both populations affected?

We thank the Reviewer for this excellent question. On reviewing our EM data, we do not have the resolution or immunogold imaging of the older highly organized mouse hearts to permit accurate quantitative comparison between SSM and IFM mitochondria. This is an important question that we should follow-up for a future careful study dedicated to this issue, and appreciate the Reviewer’s insight in pointing out this direction.

3. The 20 kDa peptide contains some TM sequence, so why is it not targeted to the PM? Is there a MTS? A schematic earlier in the text would help orient the reader – similar to what has been shown in previous articles.

We appreciate this important question from the Reviewer. By an in silico search, we could not identify a MTS in GJA1-20k. There are reports that Hsp90 helps target Cx43 to mitochondrial membrane (Rodriguez-Sinovas *et al.*, *Circ Res.*, 2006; and Rodriguez-Sinovas *et al.*, *Biochim Biophys Acta Biomem.*, 2018), which would be consistent with GJA-20k localizing to mitochondria during stress. As we have emphasized earlier, prior reports of Cx43 localization by immunocytochemistry may actually have identified GJA1-20k rather than full length Cx43 since the epitope of most anti-Cx43 antibodies is on the C-terminus and will identify GJA1-20k as well. Because GJA1-20k is hydrophilic and does not exist as membrane bound hexamers like Cx43, GJA1-20k does not require vesicle based delivery to the plasma membrane, and therefore is more available for attachment to intracellular organelles such as mitochondria.

We apologize, but need clarification on the Reviewer’s request for a schematic earlier in the text. Should our overall schematic (Figure 6 in the revised manuscript) appear earlier or does the Reviewer suggest a GJA1-20k specific explanatory schematic and/or sequence that is introduced earlier on so readers can differentiate GJA1-20k from full length Cx43 hemichannels?

4. No change in S616 phosphorylation, but why not check S637? This site is thought to be the major driver of Drp1 activation/inactivation. So changes in the phosphorylation status at 637 is critical to provide a complete evaluation of Drp1 PTM regulation.

We appreciate Reviewer #3’s thoughtful comment. We additionally analyzed phospho-DRP1 (S637) expression and found no change in the expression (Figure 2 A, B in revised manuscript).

The revised manuscript has been modified to include the above new data (Figure 2A, B):

—Results section

“DRP1 is potentiated by phosphorylation of its Serine 616 and 637 (Sabouny and Shutt 2020). We tested abundance of total DRP1 and phosphorylated DRP1 in GJA1-20k transfected HEK293 cells and found no significant difference in either the protein levels of total DRP1, DRP1-pS616, -pS637, or in the ratio (pS616/total) (Figure 2A, B). The levels of MFN1 and MFN2, and mitochondrial marker protein TOM20, a marker of cellular mitochondrial content, were also unchanged (Figure 2A, B).”

5. Is Drp1 localization altered when GJA1-20k is overexpressed? Hard to tell from Supplemental Figures No quantification.

We appreciate Reviewer #3’s valuable question. To address this issue, we quantified amount of mitochondria associated with DRP1. As shown in Figure 2—figure supplement 1C, there was no significant difference in DRP1 localization to mitochondria between GST control and GJA1-20k transfected group.

The revised manuscript has been modified to include the above new data (Figure 2—figure supplement 1C).

6. In Figure 5, the Seahorse data show a higher level of non-mitochondrial respiration in the animal samples. This difference also mutes the differences in Basal Respiration, Leak and Maximal Respiration. There is definitely a trend in the data, but the differences are not as convincing if the non-mitochondrial respiration is taken into account (unless I am missing something in the way that the data is presented).

We would like to thank Reviewer #3 for pointing this out. As the reviewer indicates, non-mitochondrial respiration is increased in GJA1 mutant mouse hearts. In the revised manuscript we provide bar graphs indicating each OCR parameter (Figure 5 —figure supplement 1C). To calculate these parameters, non-mitochondrial respiration (third measurement point of R/A treatment) was subtracted. The data confirm maximal respiration is increased in the Gja1^M213L/M213L^ mice.

To clarify this point, we have updated the Methods section:

—Methods section

“For basal respiration, we calculated the difference between the third measurement during the pre-oligomycin phase and the third measurement post-R/A. For proton leak, we calculated the difference between the third measurement during the post-oligomycin and the third measurement post-R/A. For maximum respiration, we calculated the difference between the third measurement during the post-FCCP and the third measurement post-R/A. For ATP-linked respiration, we calculated the difference between the third measurement during the pre-oligomycin phase and the third measurement post-oligomycin. For reserve capacity, we calculated the difference between the third measurement during the post-FCCP and the third measurement the pre-oligomycin phase. For non-mitochondrial respiration, we used the value at the third measurement post-R/A.”

Also, the revised manuscript has been modified to include the above new data (Figure 5C, D; Figure 5—figure supplement 1A, B):

—Results section

“Similarly, maximal respiration is increased in neonatal CMs derived from GJA1-20k deficient Gja1^M213L/M213L^ mice and maximal respiration for heterozygous Gja1^M213L/WT^ mice is between that of WT and Gja1^M213L/M213L^ (Figure 5C, D; Figure 5—figure supplement 1A, B). In addition, observing other OCR parameters, we found a decrease in ATP-linked respiration and reserve capacity in Gja1^M213L/WT^ cardiomyocytes, and an increase in proton leak and non-mitochondrial respiration in Gja1^M213L/M213L^ suggesting that there can be compensatory long-term effects of the Gja1 mutation (Figure 5—figure supplement 1C).”

7. IMO, the discussion is a little long and winding. Some of it is necessary, but I think it can be more efficiently organized to better highlight the impact of your findings.

We appreciate Reviewer #3’s thoughtful comment. In the original manuscript and especially in the revised manuscript we have included text to address potential questions from a diversity of readers, from cytoskeleton dynamics to ischemic preconditioning. We have attempted to still make the Discussion coherent but do not was to be too pithy at the expense of addressing an issue important to a particular Reviewer or reader.

8. Typo in Line 45… "stress sFriedman and".

We thank the reviewer for pointing this out. We corrected the typo.